# BRAIN-IT: IMAGE RECONSTRUCTION FROM FMRI VIA BRAIN-INTERACTION TRANSFORMER

**Roman Beliy**[*]
Department of Computer Science
Weizmann Institute of Science
Roman.beliy@weizmann.ac.il

**Amit Zalcher**[*]
Department of Computer Science
Weizmann Institute of Science

**Jonathan Kogman**
Department of Computer Science
Weizmann Institute of Science

**Navve Wasserman**
Department of Computer Science
Weizmann Institute of Science

**Michal Irani**
Department of Computer Science
Weizmann Institute of Science

## ABSTRACT

Reconstructing images seen by people from their fMRI brain recordings provides a non-invasive window into the human brain. Despite recent progress enabled by diffusion models, current methods often lack faithfulness to the actual seen images. We present "Brain-IT", a brain-inspired approach that addresses this challenge through a *Brain Interaction Transformer* (BIT), allowing effective interactions between *clusters of functionally-similar brain-voxels*. These functional-clusters are shared by all subjects, serving as building blocks for integrating information both within and across brains. All model components are shared by all clusters & subjects, allowing efficient training with a limited amount of data. To guide the image reconstruction, BIT predicts two complementary *localized* patch-level image features: (i) high-level semantic features which steer the diffusion model toward the correct semantic content of the image; and (ii) low-level structural features which help to initialize the diffusion process with the correct coarse layout of the image. BIT's design enables direct flow of information from brain-voxel clusters to localized image features. Through these principles, our method achieves image reconstructions from fMRI that faithfully reconstruct the seen images, and surpass current SotA approaches both visually and by standard objective metrics. Moreover, with only 1-hour of fMRI data from a new subject, we achieve results comparable to current methods trained on full 40-hour recordings. Project page can be found in: https://amitzalcher.github.io/Brain-IT/.

## 1 INTRODUCTION

Reconstructing visual experiences from brain activity (fMRI-to-image reconstruction) is a key challenge with broad implications for both neuroscience and brain–computer interfaces (Milekovic, 2018; Naci et al., 2012). Such reconstructions may provide a window into visual perception in the brain, enable the study of visual imagery (Cichy et al., 2012; Pearson et al., 2015), reveal dream content (Horikawa et al., 2013; Horikawa & Kamitani, 2017), and even assist in assessing disorders of consciousness (Monti et al., 2010; Owen et al., 2006). In a typical image decoding setting, subjects view natural images while their brain activity is being recorded using *functional Magnetic Resonance Imaging* (fMRI). This produces paired data of images and their corresponding fMRI scans. The task is then to reconstruct the perceived image from new (test) fMRI signals.

---

[*]Equal contribution

**(a) Reconstruction of seen images from fMRI using Brain-IT:**

**(b) Reconstruction with limited amount of training data:**

Figure 1: **Reconstruction of seen images from fMRI using "*Brain-IT*".** *(a) Image reconstructions using the full NSD dataset (40 hours per subject). (b) Efficient Transfer-learning to new subjects with very little data: Meaningful reconstructions are obtained with only 15 minutes of fMRI recordings. (Results on Subject 1)*

Early work in this domain mapped fMRI signals to handcrafted image features (Kay et al., 2008; Naselaris et al., 2009; Nishimoto et al., 2011), which were then used for image reconstruction. Subsequent studies employed deep learning methods (Beliy et al., 2019; Lin et al., 2019; Shen et al., 2019). Significant progress in reconstruction quality was achieved through the introduction of diffusion models (Chen et al., 2023; Ozcelik & VanRullen, 2023; Takagi & Nishimoto, 2023). Recent works address the scarcity of fMRI data by integrating information across multiple subjects (Scotti et al., 2024; Gong et al., 2025; Liu et al., 2025), though variability across individuals remains a challenge (Haxby et al., 2011; Yamada et al., 2015; Wasserman et al., 2024).

Despite these advances, a significant performance gap in fMRI-based image reconstruction remains. SotA methods produce reconstructions which are visually pleasing, yet often unfaithful to the actual seen image. They deviate in structural aspects (e.g., position, color), and often also miss or distort some of the semantic content. These limitations stem from the reliance on generative priors in diffusion models, which can produce realistic images even with limited guidance from brain activity. We attribute this gap to the way representations are currently extracted from fMRI, their mapping into image features, and the incorporation of these features into generative models.

In this work, we introduce ***Brain-IT***, a new approach for fMRI decoding, which produces reconstructions that more closely resemble the viewed images both structurally and semantically. At the core of our method is the "Brain Interaction Transformer" (BIT), a model that transforms brain-voxel clusters into patch-level image features. BIT's design is inspired by principles of brain organization. Neural processing is distributed across many regions rather than centralized. In the visual cortex, spatial information is organized retinotopically, preserving the layout of the visual field. Different aspects of visual representation such as color, shape, and higher-level semantic features are localized to distinct yet interconnected regions (Kandel et al., 2013). Guided by these principles, BIT clusters functionally similar brain-voxels [1] and 'summarizes' each cluster with a single Brain Token (vector). The resulting clusters (Brain Tokens) are shared across subjects, thus capturing similar roles across individuals, and support effective integration of information (both *within* and *across* brains). Brain Tokens interact to refine their representations and are mapped to localized image-feature tokens. This enables direct information flow from functional clusters to *localized* image features.

---

[1] A 'brain-voxel' is a small 3D cube of brain tissue, whose activity is estimated via fMRI and represented by a single scalar value.

Our pipeline integrates two complementary branches, both driven by image features *predicted by BIT*: (i) A high-level semantic branch, whose predicted features steer the diffusion model toward the correct semantic content of the image; and (ii) A low-level structural branch, whose VGG-predicted features are inverted through a Deep Image Prior (DIP) framework to reconstruct a coarse image layout. The low-level image establishes the global structure, while the diffusion process refines it under semantic conditioning. We adopt this design to combine diffusion's strong generative priors with VGG's structural fidelity and DIP's convolutional inductive biases. Combining BIT's novel localized feature predictions with the two branches yields reconstructions that preserve structural fidelity, semantic detail, and perceptual realism.

***Brain-IT*** outperforms previous methods on all standard metrics, resulting in reconstructions that are more faithful to the seen images. Moreover, ***Brain-IT*** supports efficient transfer learning to new subjects with little subject-specific training data: with only 1 hour of fMRI data from a new subject, it achieves performance comparable to existing methods trained on the full 40-hour dataset. This efficient transfer learning is enabled by our model's design: since the atomic training unit is a *functional cluster of brain voxels* shared across subjects, and all clusters use the same network weights, we are able to learn representations that generalize across individuals and can be adapted to new subjects with limited training data.

**Our contributions are therefore as follows:**

- We present ***Brain-IT***, a brain-inspired approach for fMRI-to-image reconstruction that faithfully reconstructs seen images, yielding SotA results (both visually and by quantitative metrics).

- We introduce a 'Brain Interaction Transformer' (BIT), which maps *functional brain-voxel clusters* to *localized image features*, allowing effective integration of information across multiple brains.

- A new approach to low-level image reconstruction from fMRI via *Deep Image Prior* (DIP), which accurately predicts the coarse image layout (forming a powerful input to the diffusion process).

- Transfer-learning to new subjects with little data: Meaningful reconstructions obtained with only 15 minutes of fMRI recordings. Results on 1 hour are comparable to prior methods on 40 hours.

## 2 RELATED WORK

fMRI-to-image reconstruction is a well-established field, seeing significant progress in the past decade. Early work mapped fMRI to handcrafted image features (Kay et al., 2008; Naselaris et al., 2009; Nishimoto et al., 2011), followed by works that mapped fMRI to deep CNN features (Güçlü & Van Gerven, 2015; Zhang et al., 2018a; Shen et al., 2019). End-to-end methods have emerged (Seeliger et al., 2018; St-Yves & Naselaris, 2018; Beliy et al., 2019), with later ones predicting latent codes of VAEs or GANs (Han et al., 2019; Lin et al., 2019; Mozafari et al., 2020; Qiao et al., 2020; Ren et al., 2021). Recently, diffusion models have improved realism and faithfulness by turning predicted features into high-quality images (Chen et al., 2023; Ozcelik & VanRullen, 2023; Takagi & Nishimoto, 2023). In parallel, the community has increasingly focused on integrating information across multiple subjects to improve generalization under limited data per subject (Scotti et al., 2024; Gong et al., 2025; Huo et al., 2024; Xia et al., 2024b; Ferrante et al., 2024; Liu et al., 2025; Shen et al., 2024). Next, we review related efforts along 3 key axes: predicting image features from fMRI, leveraging cross-subject information, and predicting intermediate low-level images.

**Image Features Prediction.** Current methods predict image features from fMRI, such as semantic CLIP or VAE latent embeddings. Most use simple linear models or MLPs (Takagi & Nishimoto, 2023; Xia et al., 2024a; Wang et al., 2024b); others build on top with unCLIP diffusion priors (Gong et al., 2025; Scotti et al., 2023). A key limitation of these methods is that they typically compress all voxels into a single global fMRI embedding via a fully connected layer before predicting image features. Since visual information is distributed across multiple distinct yet interconnected brain regions, such a projection fails to fully exploit this distributed nature of the brain. Two recent works introduce spatial voxel groupings (Huo et al., 2024; Shen et al., 2024), which use voxel patches in shared anatomical space. However, they predict a single *global* image representation, making it difficult to reconstruct localized image information. Our approach addresses these issues by forming functionally shared voxel clusters which are mapped directly to localized image features, avoiding projection to a global fMRI embedding and yielding more accurate image-feature predictions.

**Cross-Subject Integration.** Few prior works support multi-subject training (Lin et al., 2022; Ferrante et al., 2024; Gong et al., 2025; Huo et al., 2024; Scotti et al., 2024). A common aspect of these

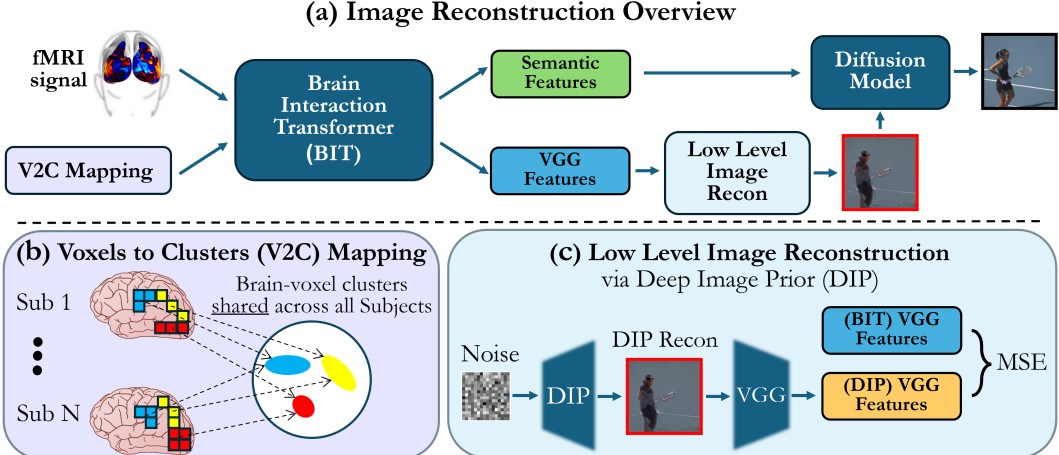

Figure 2: **Overview of *Brain-IT*.** **(a) Brain Interaction Transformer (BIT)** *transforms fMRI signals into* Semantic *and* VGG *features using the Voxel-to-Cluster (V2C) mapping. Two branches are applied: (i) the Low-Level branch reconstructs a coarse image from VGG features, used to initialize the (ii) Semantic branch, which uses semantic features to guide the diffusion model.* **(b) Voxel-to-Cluster mapping (V2C):** *each voxel from every subject is mapped to a functional cluster shared across subjects.* **(c) Low-level branch:** *VGG-predicted features are inverted using Deep Image Prior (DIP) to reconstruct a coarse image layout.*

methods is that alignment is predominantly fMRI-centric: they treat each fMRI scan as a single entity and rely on shared embeddings of entire scans across subjects. As a result, they can only exploit shared representations at the scan level, overlooking the more frequent similarities that exist between individual brain-voxels, both within a single subject and across different subjects. Inspired by the multi-subject *Image-to-fMRI Encoder* of (Beliy et al., 2024), we adopt a voxel-centric model that shares network weights across all brain-voxels, both within and across subjects. By sharing most model components and leveraging voxel-level similarities, our approach integrates data effectively across subjects and adapts efficiently to new ones even with limited data.

**Low-Level Image Reconstruction.** Most existing methods regress directly from fMRI to the latent space of the diffusion model (e.g. VAE latent space) using MLPs or linear layers (Scotti et al., 2024; Gong et al., 2025; Ozcelik & VanRullen, 2023). Alternatively, NeuroPictor (Huo et al., 2024) manipulates internal feature maps within the U-Net of Stable Diffusion using features predicted from fMRI. Unlike these approaches, our method introduces a complementary low-level branch that predicts VGG-based representations inspired by LPIPS (Simonyan & Zisserman, 2014; Zhang et al., 2018b) and inverts them through a Deep Image Prior (DIP) framework. This setup results in more faithful low-level predictions, reconstructing global structure and low-level visual features.

## 3 BRAIN-IT PIPELINE

Our ***Brain-IT*** pipeline reconstructs images seen by subjects directly from their fMRI activations. This section provides a full explanation of the approach (Fig. 2a), while the technical details and architecture of the BIT model are described in Sec. 4. The pipeline consists of two main stages: (i) *Image Feature Prediction:* The Brain Interaction Transformer (BIT) maps fMRI signals into meaningful visual features using a shared Voxel-to-Cluster mapping (Fig. 2b). Two BIT models are trained: one predicts adapted CLIP embeddings (semantic features), and the other predicts VGG features (low-level). (ii) *Image Reconstruction:* The predicted features are used in two branches. VGG features are inverted through a Deep Image Prior (DIP) framework to produce a coarse reconstruction (Fig. 2c). Semantic features condition a diffusion model to generate an image aligned with the fMRI signal. At inference, the DIP-based coarse reconstruction initializes the diffusion process, which is then refined under semantic guidance to produce a detailed, faithful image.

### 3.1 IMAGE FEATURE PREDICTION

Given an fMRI recording of a subject viewing an image, the goal is to extract visual information by predicting image features that guide reconstruction. This stage has two main components: the *Voxel-to-Cluster Mapping*, and the *Brain Interaction Transformer*.

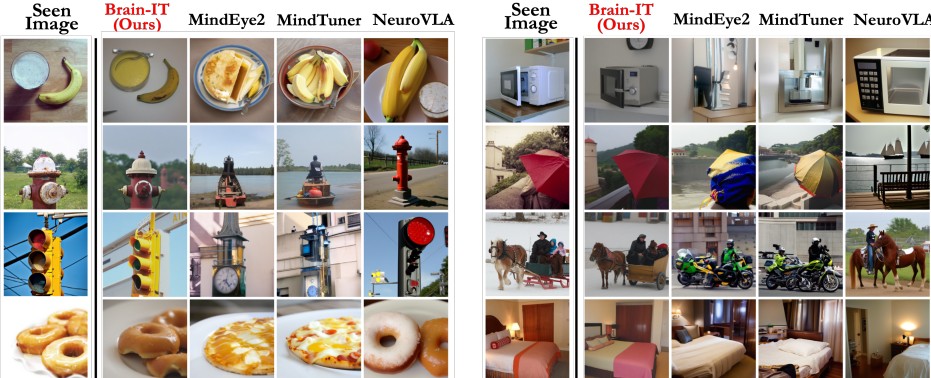

Figure 3: **Comparing methods on 40-hour data (for Subject 1).** *Brain-IT is compared to 3 leading methods, yielding reconstructions that better preserve both semantic content and low-level visual properties. **Brain-IT** better reconstructs the correct objects with relevant structural details (e.g., orientation, color), providing reconstructions more faithful to the seen images. See many more examples in Appendix Fig. S2*

**Voxels-to-Clusters Mapping (V2C).**   We start by mapping *functionally similar brain-voxels* from all brains into few (128) shared clusters (Fig. 2.b). Each cluster captures similar roles across all subjects/voxels, thus supporting effective integration of information both *within* and *across* brains. This Voxels-to-Clusters (V2C) mapping assigns each voxel to a single functional cluster, reducing 40K voxels into 128 clusters. We determine the functional similarity of brain-voxels using the Encoding "voxel embeddings" obtained by the *Brain Encoder* of Beliy et al. (2024), which implicitly learns an individual embedding for each brain-voxel (for any subject). These encoded voxel embeddings capture the functional role of each brain-voxel, for example, whether it responds to low-level or semantic (high-level) visual features, whether it attends to specific spatial locations in an image, or to the entire image globally, etc. To map voxels from all subjects into clusters, we apply a Gaussian Mixture Model (GMM) on the voxel embeddings.

**Brain Interaction Transformer (BIT).**   The BIT model takes as input the fMRI signals together with the V2C mapping and predicts image features (e.g., adapted CLIP embeddings for the semantic branch, VGG features for the low-level branch). First, BIT transforms brain activations into Brain Tokens, each summarizing the information from the voxels within a voxel cluster. This process incorporates two learnable embeddings to integrate brain information: a per-voxel embedding, which captures each voxel's functionality, and a per-cluster embedding, which captures each cluster's overall functionality. This serves as an information-selection bottleneck by determining how voxel information is aggregated into Brain Tokens. The resulting Brain Tokens are then refined via attention layers. Self-attention layers model relationships across Brain Tokens (voxel clusters), while cross-attention layers extract specific image feature information from them. This enables direct information flow from Brain Tokens to predict localized image features as demonstrated in Appendix Sec. C.4, by attention map visualization.

**Enriching the Training Data.**   The training is performed using pairs of fMRI measurements and the corresponding images viewed by the subject, taken from the NSD dataset (see Sec. 5.1). The model is trained on data from all subjects simultaneously. Since the subject-specific data is limited, we further enrich the training set with **"external images"** (natural images without any fMRI). Specifically, we use ∼120k natural images from the unlabeled portion of COCO dataset. To incorporate these images, we employ the Image-to-fMRI Encoder of Beliy et al. (2024), which predicts fMRI responses for the external images, thereby creating additional training pairs. Such training is particularly important in Transfer-Learning to new subjects, with very little subject-specific data.

## 3.2   IMAGE RECONSTRUCTION

Our image reconstruction consists of 2 complementary branches: ***Semantic Image Generation*** that conditions diffusion model on adapted-CLIP embeddings to generate a semantic image, and ***Low-level Image Reconstruction***  that inverts VGG features via Deep Image Prior (DIP) to produce a coarse image. At inference, we use ***Dual-Branch Generation***: the low-level branch provides a coarse initialization, and the diffusion model refines it under semantic guidance.

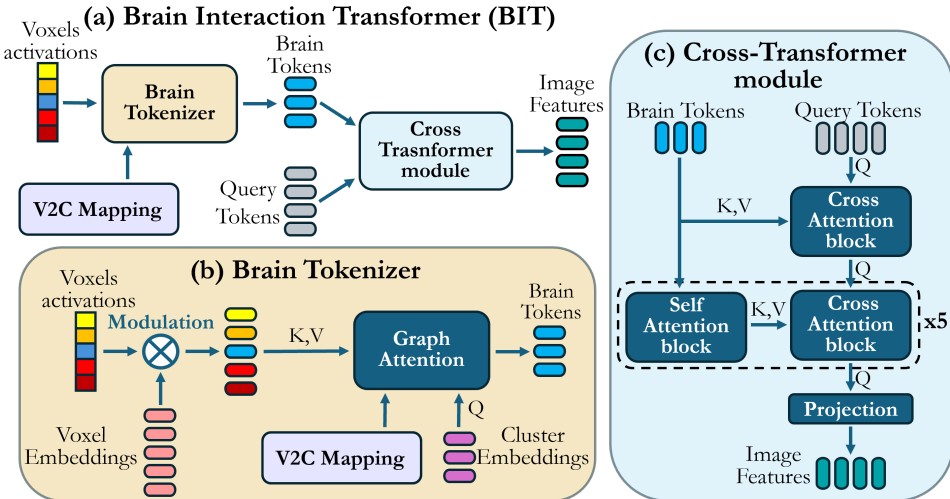

Figure 4: **Architecture of the Brain-Interaction-Transformer (BIT).** (Sec. 4)

**Semantic Image Generation.** Our semantic branch (Fig. 2.a) aims to generate images which are simultaneously consistent with the distribution of natural images, and semantically faithful to the underlying seen image. It leverages the strong prior of diffusion models together with the adapted CLIP image embeddings predicted by BIT, to guide generation toward the seen image. We use the diffusion model introduced by Scotti et al. (2024), which follows an unCLIP-style approach: an adaptation of Stable Diffusion XL (SDXL) modified to condition on all 256 spatial OpenCLIP ViT-bigG/14 tokens, enabling high-fidelity image reconstruction from CLIP tokens. Our training procedure consists of two stages. (i) Feature Alignment: We first train the BIT model to predict the 256 spatial tokens of OpenCLIP ViT-bigG/14 . This step aligns BIT's outputs with the conditioning signal expected by the diffusion model. Training uses paired fMRI activations and image CLIP embeddings with an L2 loss. (ii) Joint Training: In the second stage, we jointly train BIT and the diffusion model. BIT's predicted tokens are used to condition the image diffusion process. The two models are optimized together with the standard diffusion loss. Noise is added to the target image, and the diffusion process is trained to denoise it. Importantly, the BIT output representation does not need to remain aligned with the original CLIP embeddings. The two networks can establish a representation that is better suited for conditioning image generation from fMRI activations.

**Low level Image Reconstruction.** Our low-level branch (Fig. 2.c) aims to reconstruct the coarse structural layout and low-level visual features of the seen image, and is used to initialize the semantic-based diffusion process. Treating VGG features as a robust intermediate representation for low-level image reconstruction, we train a BIT model to predict multi-layer VGG features using the InfoNCE (van den Oord et al., 2018) loss, with each position in each layer predicted as a separate token. At inference time, these predicted features are inverted through a Deep Image Prior (DIP) (Ulyanov et al., 2018) to generate a low level image. DIP provides a strong image prior through the inductive bias of convolutional neural networks. The DIP model is optimized to reconstruct an image whose VGG features are as close as possible to the BIT-predicted VGG features (see (Fig. 2.c). To achieve this, the DIP generated image is passed through a frozen VGG network, whose activations are optimized to match the BIT-predicted features under an L2 loss (details in Sec. D.2).

**Dual-Branch Generation** At inference time, the 2 branches are combined: the low-level image initializes the diffusion process, providing coarse image structure, which is then refined into detailed reconstructions with semantic guidance. Reconstructions guided by semantic embeddings alone tend to be semantically consistent but not structurally faithful to the visual stimulus, often failing to preserve low-level details of the seen image such as colors and structure. Kamb & Ganguli (2025) demonstrated that diffusion models generate images in a coarse-to-fine manner: first establishing a global layout and then progressively refining details. Building on this insight, we initialize the diffusion process with a noisy version of the predicted low-level image, which supplies reliable structure. The diffusion model then updates this initialization under semantic conditioning, yielding reconstructions that capture both structure and semantic details, and are more faithful to the seen image. Finally, we apply a refinement stage as in Scotti et al. (2024) in which the image is passed

| Method | Low-Level | | | | High-Level | | | |
|---|---|---|---|---|---|---|---|---|
| | PixCorr ↑ | SSIM ↑ | Alex(2) ↑ | Alex(5) ↑ | Incep ↑ | CLIP ↑ | Eff ↓ | SwAV ↓ |
| *MindEye* (Scotti et al., 2023) | 0.319 | 0.360 | 92.8% | 96.9% | 94.6% | 93.3% | 0.648 | 0.377 |
| *Brain-Diffuser* (Ozcelik & VanRullen, 2023) | 0.273 | 0.365 | 94.4% | 96.6% | 91.3% | 90.9% | 0.728 | 0.421 |
| Takagi & Nishimoto (2023) | 0.246 | 0.410 | 78.9% | 85.6% | 83.8% | 82.1% | 0.811 | 0.504 |
| *DREAM* (Xia et al., 2024a) | 0.274 | 0.328 | 93.9% | 96.7% | 93.4% | 94.1% | 0.645 | 0.418 |
| *UMBRAE* (Xia et al., 2024b) | 0.283 | 0.341 | 95.5% | 97.0% | 91.7% | 93.5% | 0.700 | 0.393 |
| *NeuroVLA* (Shen et al., 2024) | 0.265 | 0.357 | 93.1% | 97.1% | 96.8% | **97.5%** | 0.633 | 0.321 |
| *MindBridge* (Wang et al., 2024a) | 0.151 | 0.263 | 87.7% | 95.5% | 92.4% | 94.7% | 0.712 | 0.418 |
| *NeuroPictor* (Huo et al., 2024) | 0.229 | 0.375 | 96.5% | 98.4% | 94.5% | 93.3% | 0.639 | 0.350 |
| *MindEye2* (Scotti et al., 2024) | 0.322 | 0.431 | 96.1% | 98.6% | 95.4% | 93.0% | 0.619 | 0.344 |
| *MindTuner* (Gong et al., 2025) | 0.322 | 0.421 | 95.8% | 98.8% | 95.6% | 93.8% | 0.612 | 0.340 |
| ***BrainIT (Ours)*** | **0.386** | **0.486** | **98.4%** | **99.5%** | **97.3%** | 96.4% | **0.564** | **0.320** |
| **Results on 1 hour (out of 40)** | | | | | | | | |
| MindEye2 (1 hour) | 0.195 | 0.419 | 84.2% | 90.6% | 81.2% | 79.2% | 0.810 | 0.468 |
| MindTuner (1 hour) | 0.224 | 0.420 | 87.8% | 93.6% | 84.8% | 83.5% | 0.780 | 0.440 |
| ***BrainIT (1 hour)*** | **0.331** | **0.473** | **97.1%** | **98.6%** | **94.4%** | **93.0%** | **0.648** | **0.370** |

Table 1: **Evaluation of different methods.** *We report low- and high-level metrics comparing Brain-IT with other reconstruction methods.* **Top:** *methods trained on the full 40 hours data.* **Bottom:** *results with only 1 hour of subject-specific data. All results are averaged across Subjects 1,2,5,7 from NSD.* **Brain-IT** *outperforms all baselines in 7 of 8 metrics, demonstrating strong semantic fidelity and structural accuracy. Importantly, with just 1 hour of data,* **Brain-IT** *is comparable to prior methods trained on the full 40 hours.*

through a pre-trained SDXL diffusion model without any text prompt, producing visually enhanced reconstructions (see Sec. A.4).

# 4 BRAIN INTERACTION TRANSFORMER (BIT)

The BIT model predicts image features from voxel activations (fMRI) (see Fig. 4.a). The **Brain Tokenizer** maps the fMRI activations into Brain Tokens, which are representations of the aggregated information from all the voxels of a single cluster (one token per cluster). The **Cross-Transformer Module** integrates information from the Brain Tokens to refine their representation, and employs query tokens to retrieve information from the Brain Tokens and transform it into image features, with each query token predicting a single output image feature.

**Brain Tokenizer:** The Brain Tokenizer (Fig. 4.b) transforms fMRI brain activations into informative *Brain Tokens*, where each Brain-Token 'summarizes' the information of a single functional cluster of brain-voxels. The architecture of the Tokenizer detailed in Fig. 4.b is explained next. It incorporates two learnable embeddings, initialized randomly and optimized during training: (i) *Voxel Embedding*: a 512-dimensional vector, similar to Beliy et al. (2024), trained to capture each voxel's functionality. Importantly, while the voxel embedding of Beliy et al. (2024) were optimized for image-to-fMRI Encoding, our voxels embeddings are optimized for fMRI-to-Image Decoding (which results in very different voxel embeddings). Each voxel embedding (vector) modulates the fMRI activation (scalar) by multiplying the voxel activation with its corresponding Voxel Embedding. (ii) *Cluster Embedding*: a 512-dimensional vector for each cluster of voxels, trained to capture the cluster's overall functionality. This serves as an information-selection bottleneck by determining how voxel information is aggregated, and is shared by all subjects. The Brain-Tokenizer forms Brain Tokens by combining modulated voxel activations (voxel activations multiplied by their voxel embeddings) within each functional cluster. This is implemented with a single-head graph attention layer (Shi et al., 2021), which restricts attention to act between each voxel and its corresponding Cluster Embedding according to the Voxels-to-Cluster (V2C) mapping. The modulated voxel activations are used as keys (K) and values (V), while the learned Cluster Embeddings act as queries (Q). The output is a set of 512-dimensional Brain Tokens, each representing one cluster.

**Cross-Transformer:** The Cross-Transformer Module (see Fig. 4.c) refines the representation of Brain Tokens through their interactions. It integrates their information via query tokens to predict image features, with each query token corresponding to a single output image feature (architecture

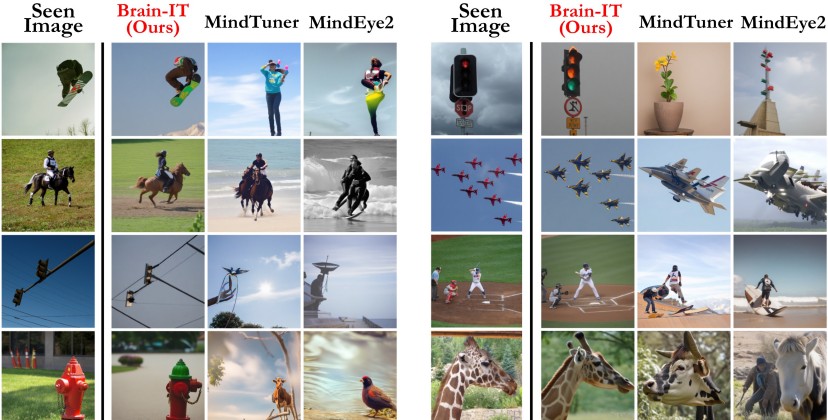

Figure 5: **Reconstruction with limited amount of subject-specific data (1 hour).** *We compare **Brain-IT** against 2 leading approaches which provide also 1-hour reconstructions (MindEye2 & MindTuner) for Subj1. **Brain-IT** demonstrates greater fidelity to the seen image. See many more examples in App.Fig. S3*

adapted from (Fu et al., 2024)). Self-attention layers model interactions across voxel clusters, while cross-attention layers enable direct information flow from Brain Tokens to predict localized image features. In the first cross-attention block, query tokens (initialized randomly and optimized during training) attend to the Brain-Tokens, which serve as keys (K) and values (V). This is followed by five cross-transformer blocks, each consisting of a self-attention layer that refines the Brain-Token representations, and a cross-attention layer that refines the predicted image features. Both self- and cross-attention layers contain eight heads. A final projection layer after the last cross-attention block maps the representation to the dimensionality of the output image features.

## 5 EXPERIMENTAL RESULTS

This section presents the effectiveness of **Brain-IT** for reconstructing images from brain signals (fMRI). We first briefly describing the experimental framework (Sec. 5.1). We then show quantitative and qualitative results from our full pipeline, showing that it outperforms current SotA methods (Sec. 5.2). Next, we evaluate the transfer learning capabilities of **Brain-IT**, showing that training on a new subject with only 1 hour of data produces high-quality results, comparable to previous methods trained on the full 40 hours, and significantly better than ones trained on 1 hour (Sec. 5.3). Additionally, we demonstrate the robustness of our model through qualitative evaluation of image reconstructions on "*NSD Synthetic*" Gifford et al. (2024), which is an *out-of-distribution* dataset (Fig. S7)

### 5.1 EXPERIMENTAL SETTING

**Dataset.** We used the Natural Scenes Dataset (NSD) (Allen et al., 2022), a large publicly available 7-Tesla fMRI dataset that records fMRI of 8 subjects as they viewed diverse images drawn from COCO (Lin et al., 2014). The dataset contains ~73,000 Image-fMRI pairs, comprising ~9,000 unique images per subject and 1,000 images shared by all subjects. As in prior NSD studies, we used the shared images as the test-set. For voxel selection, we adopted the post-processed version provided by Gifford et al. (2023), which includes ~40k voxels, mainly vision-related cortical areas.

**Evaluation Metrics.** We follow the standard practice of evaluating model performance using both low- and high-level image metrics. Unless stated otherwise, our tables display averaged results across four subjects from the NSD dataset (Subjects 1,2,5,7), to be compatible with previous methods. *PixCorr* denotes the pixel-wise correlation between ground-truth and reconstructed images, while *SSIM* refers to the (grayscale) structural similarity index (Wang et al., 2004) *Alex(2)*, *Alex(5)*, *Incep*, *CLIP*, measure two-way identification accuracy based on correlations between image and brain embeddings at different feature levels. Finally, *EfficientNet-B1* (*"Eff"*) (Tan, 2019) and *SwAV-ResNet50* (*"SwAV"*) (Caron et al., 2020) report average correlation distance, where lower values indicate better performance. Moreover, in Table T6 we evaluate our reconstructions on 3 additional metrics: *1000-way CLIP* retrieval, *LPIPS* (Zhang et al., 2018b), and *Color-SSIM*.

### 5.2 RECONSTRUCTION RESULTS

We compare both quantitative and qualitative image reconstructions results to previous prominent methods. In Table 1, we report both low-level and high-level quantitative evaluation metrics. Across

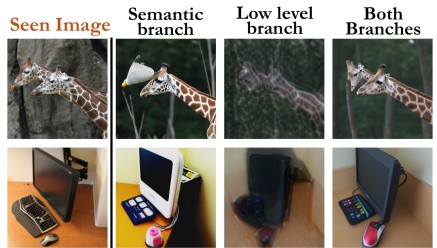

Figure 6: **Two-Branch Contribution Reconstructions.** *Sample results of the semantic branch, low-level branch, and both combined.*

Table 2: **Branch contributions.** *Quantitative results for each reconstruction branch alone (semantic & low-level), and both combined.*

| Method | Low-Level Metrics | | High-Level Metrics | |
|---|---|---|---|---|
| | SSIM ↑ | Alex(5) ↑ | Incep ↑ | CLIP ↑ |
| Low-level branch | **0.505** | 99.1% | 95.9% | 85.8% |
| Semantic branch | 0.431 | 99.4% | 96.1% | 95.2% |
| Combined | 0.486 | **99.5%** | **97.3%** | **96.4%** |

all 4 low-level metrics, **Brain-IT** surpasses all previous approaches by a large margin. We attribute this to the combination of our high-quality BIT predictions of low-level VGG features, along with their DIP inversion pipeline. For high-level metrics, our method achieves the best performance in 3 metrics and ranks second in one, where NeuroVLA performs better. Overall, our method demonstrates strong performance across both high-level semantic and low-level structural fidelity.

Fig. 3 provides visual comparisons of our method to three prominent approaches. Many more examples and comparisons to additional methods can be found in Fig. S2 and reconstructions from other subjects are shown in Fig. S4. **Brain-IT** achieves significantly higher fidelity to the seen image than prior methods, both semantically and structurally (e.g., the correct objects, their spatial arrangements, color and shape). Despite the inherent resolution limits of fMRI, our model successfully reconstructs most of the relevant features of the original images.

## 5.3 TRANSFER LEARNING

**Brain-IT** facilitates highly efficient transfer learning to new subjects with very little subject-specific data. This is particularly important since collecting fMRI recordings is costly and time-consuming. All model components are shared by all subjects, with the exception of the voxel embeddings which are subject-specific. These per-voxel vectors (which are part of the Brain Tokenizer), are optimized to capture the functionality of each voxel and to integrate information for predicting image features from this voxel's fMRI activation. Adapting to a new subject therefore requires optimizing only the voxel embeddings, whose expressivity is constrained by the frozen network components. This enables efficient adaptation even with very little training data. For details, see Sec. D.4.

Table 1 (bottom) shows quantitative comparison of **Brain-IT** against MindEye2 and MindTuner, all trained with only 1 hour of data from a new subject. **Brain-IT** performs *significantly* better on all metrics. Note that our model shows a larger performance gap over competitors with 1 hour of data than with 40 hours. Importantly, with just 1 hour of data, our results are comparable to previous methods trained on the full 40-hour dataset, highlighting the strong efficiency of our approach. These findings are further visually illustrated in Fig. 5, where our 1-hour reconstructions surpass MindEye2 and MindTuner trained on the same duration. Many more examples, as well as direct visual comparison between 1-hour and 40-hours reconstructions, are provided in Fig. S3. We further demonstrate that our method maintains strong performance with only 30 minutes or even 15 minutes of data. Visual examples are shown in Fig. 1 (bottom) and Fig. S3, with quantitative results provided in Table T7. To our best knowledge, this is the first demonstration of high-quality reconstructions from as little as 15 minutes of fMRI data, demonstrating the generalization power of our method.

## 6 ABLATIONS AND ANALYSIS

This section evaluates the contribution of the two branches (semantic and low-level) and provides additional analysis. Further ablations of other components and factors are provided in Sec. A. Those include contribution of: (i) **External data**, incorporating unlabeled images with no fMRI into training (Sec. A.1); (ii) **Functional vs. Anatomical clustering**, comparing voxel clusters based on functional representations (as used in Brain-IT) vs. anatomical based clustering (Sec. A.2); (iii) **Number of clusters**, examining the effect of the number of clusters on reconstruction performance (Sec. A.3). We also provide an interpretability analysis of *Brain-IT* via BIT cross-attention maps, showing spatial and semantic selectivity of brain tokens.

**Two-Branch Contribution:** Fig. 6 shows sample reconstructions using the semantic branch alone, the low-level branch alone (via DIP without diffusion), and both branches together (more examples

in Fig. S8). The semantic branch captures the semantic content of the image, while the low-level branch preserves coarse outlines and structural detail. The low-level branch provides strong structural cues: in the computer image it captures the colors of the screen and desk, and in the giraffe image it recovers the outlines of the two giraffes. When combined, the two branches yield reconstructions that are more semantically accurate and structurally faithful. Table 2 reports selected quantitative metrics (full results in Table T4) consistent with these observations: the low-level branch achieves higher scores on structural fidelity metrics such as pixel correlation and SSIM, while the semantic branch excels on perceptual metrics. Notably, the combined reconstructions outperform both individual branches on several metrics, confirming their complementary contributions.

**Brain-token selectivity.** To better understand how Brain-IT uses fMRI information, we analyze the BIT cross-attention maps between brain tokens (voxel clusters) and query tokens (image-feature positions). Averaging attention across layers and heads reveals that different brain tokens consistently contribute to specific spatial locations in the predicted feature maps, exhibiting a clear contralateral organization. Beyond spatial effects, some tokens preferentially attend to regions associated with particular semantic concepts (e.g., faces, limbs, text), indicating semantic selectivity. Full visualizations are provided in Appendix C.4 (Figs. S5 and S6).

## 7 DISCUSSION

We introduce **_Brain-IT_**, a brain-inspired framework for reconstructing images from fMRI. It enables direct flow of information from _functional clusters_ of brain-voxels, to localized image features (both low-level structural image features, and high-level semantic ones). The shared functional clusters allow effective integration of information across multiple brains, and efficient Transfer-Learning to new brains, yielding high-quality reconstructions with very little subject-specific training data. **_Brain-IT_** overcomes key limitations of prior approaches, providing high-quality fMRI-to-Image reconstruction, with significantly better structural and semantic similarity to the real seen image. Despite these advances, reconstructions remain imperfect, with semantics and fine-grained details sometimes inaccurate (see Appendix Fig. S9 for failure examples). While this may reflect limitations of the fMRI signal itself, future work could explore more expressive feature spaces where current method fails. On a different frontier – our Brain-Transformer (BIT), which enables direct interactions between different functional brain sub-regions, may be applied to other neuroscience applications. By analyzing the information flow between functional sub-regions, the model may help reveal what is represented where in the brain, how inter-regional interactions support visual processing, and how these relate to perception. Our method's strong Transfer-Learning capabilities may further make it suitable for more diverse neural signals and datasets, allowing a pretrained model to be adapted to new studies with minimal data to answer task-specific neuroscientific questions.

## 8 ACKNOWLEDGMENTS

This research was funded by the European Union (ERC grant No. 101142115).

# I. ETHICS STATEMENT

We use only publicly available datasets where all the necessary measures to ensure ethical conduct were made. Additionally this work is a continuation of line of works on brain decoding and is a natural extension of previous works.

# II. REPRODUCIBILITY STATEMENT

We provide the details of our method in the main paper and additional technical information in Sec. D. Upon publication, we will release our full code with saved checkpoints to ensure that all results are easily reproducible. We will also share all reconstructed images, enabling researchers to easily compare our method under new evaluation metrics.

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

# Appendix

The appendix is organized into four sections. Sec. A presents ablations and analysis, while Sec. B and Sec. C provide further results. Specifically, Sec. B reports additional evaluation metrics for our reconstructions, and Sec. C includes many more visual reconstructions, showcasing additional images and comparisons for both the full training and transfer learning pipelines. Finally, Sec. D describes further technical details.

## A    ABLATIONS AND ANALYSIS

We describe additional experiments analyzing different components and factors of our method. We begin with an analysis of the contribution of **External Images**, followed by an evaluation of different **Clustering Methods**, and finally examine the effect of the **Number of Clusters**.

### A.1    INFLUENCE OF UNLABELED IMAGES

To further assess the impact of enlarging the training dataset, we evaluate the role of external images in our framework. As described in Sec. 3, training is based on paired fMRI-image samples from the NSD dataset. To mitigate the limited subject-specific data, we additionally use ∼120k natural images from the unlabeled portion of COCO dataset with predicted fMRI responses generated by the Image-to-fMRI encoder of Beliy et al. (2024). As shown in Table T1, incorporating external images provides improvements across most evaluation metrics, further supporting our choice to enlarge the training data in this way.

| Method | Low-Level Metrics | | | | High-Level Metrics | | | |
|---|---|---|---|---|---|---|---|---|
| | PixCorr ↑ | SSIM ↑ | Alex(2) ↑ | Alex(5) ↑ | Incep ↑ | CLIP ↑ | Eff ↓ | SwAV ↓ |
| W/o External Images | 0.365 | **0.494** | 97.5% | 99.1% | 96.5% | 95.2% | 0.592 | 0.340 |
| With External Images | **0.386** | 0.486 | **98.4%** | **99.5%** | **97.3%** | **96.4%** | **0.564** | **0.320** |

Table T1: **Ablation study on the effect of using external images during training.** Incorporating predicted fMRI responses improves performance on almost all evaluation metrics, highlighting the benefit of augmenting the limited subject-specific training data with additional unlabeled images.

### A.2    FUNCTIONAL VS. ANATOMICAL CLUSTERING

Our method relies on functional clustering, assigning voxels to clusters based on encoding embeddings from the Universal Encoder. As an anatomical alternative, we aligned all subjects to FSaverage space and clustered voxels using either (i) spatial 3D coordinates or (ii) Schaefer atlas–based cortical parcellations (e.g., Schaefer 7 networks Schaefer et al. (2018)). For the subset of voxels used here, the Schaefer-1000 atlas yielded 198 clusters, and the Schaefer-400 atlas yielded 84 clusters. This demonstrates our framework's compatibility with multiple clustering strategies, though functional clustering provides superior reconstruction performance.

| Method | Low-Level | | | | High-Level | | | |
|---|---|---|---|---|---|---|---|---|
| | PixCorr ↑ | SSIM ↑ | Alex(2) ↑ | Alex(5) ↑ | Incep ↑ | CLIP ↑ | Eff ↓ | SwAV ↓ |
| 3D-coordinates clustering | 0.378 | **0.491** | 98.1% | 99.3% | 96.6% | 95.6% | 0.589 | 0.336 |
| Schaefer-400 parcellation | 0.371 | 0.477 | 98.3% | 99.3% | 97.0% | 96.1% | 0.576 | 0.333 |
| Schaefer-1000 parcellation | 0.378 | 0.475 | **98.7%** | 99.4% | **97.4%** | 96.3% | 0.572 | 0.331 |
| Functional clustering | **0.386** | 0.486 | 98.4% | **99.5%** | 97.3% | **96.4%** | **0.564** | **0.320** |

Table T2: **Quantitative comparison Functional vs Anatomical clustering.** Results average across subjects 1, 2, 5, and 7 from the Natural Scenes Dataset.

### A.3  EFFECT OF CLUSTERS NUMBER

The number of clusters is a hyperparameter in our method. Here we show how varying this number affects performance. The results indicate that our approach is largely robust to this choice: performance remains strong across a wide range, provided the cluster count is not too small.

| # Clusters | Low-Level | | | | High-Level | | | |
|---|---|---|---|---|---|---|---|---|
| | PixCorr ↑ | SSIM ↑ | Alex(2) ↑ | Alex(5) ↑ | Incep ↑ | CLIP ↑ | Eff ↓ | SwAV ↓ |
| 8 | 0.379 | 0.492 | 97.6% | 99.1% | 96.6% | 95.6% | 0.588 | 0.334 |
| 16 | 0.380 | 0.491 | 97.8% | 99.3% | 96.6% | 95.6% | 0.582 | 0.334 |
| 32 | 0.383 | 0.486 | 98.2% | 99.3% | 97.0% | 96.0% | 0.576 | 0.325 |
| 64 | 0.385 | 0.486 | 98.3% | 99.5% | 97.2% | 96.1% | 0.569 | 0.321 |
| 128 | 0.386 | 0.486 | 98.4% | 99.5% | 97.3% | 96.4% | 0.564 | 0.320 |
| 256 | 0.391 | 0.487 | 98.6% | 99.5% | 97.6% | 96.3% | 0.565 | 0.321 |
| 512 | 0.392 | 0.491 | 98.5% | 99.5% | 97.4% | 96.1% | 0.569 | 0.322 |

Table T3: **Quantitative comparison across cluster numbers.** The model is relatively robust to the number of voxel clusters, with a slight decrease in performance for small numbers of cluster.

### A.4  REFINEMENT STAGE VISUAL RESULTS

we apply a refinement stage as in Scotti et al. (2024) in which the image is passed through a pre-trained SDXL diffusion model. This refinement improves overall realism, smooths textures, and corrects facial details, producing visually enhanced reconstructions. The figure shows that the additional refinement slightly improves perceptual quality of the generated images.

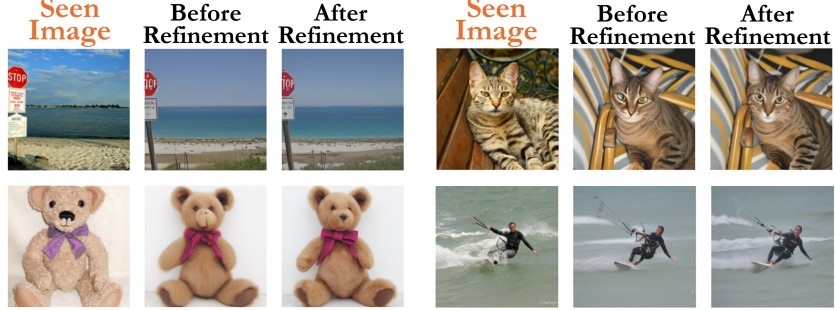

Figure S1: **Refinement stage using a pre-trained SDXL diffusion model.** The SDXL model enhances global realism, smooths textures, and corrects fine details such as the cat eyes, resulting in more natural and visually coherent reconstructions.

## A.5 Two-Branch Contribution

We evaluate the semantic and low-level branches individually and together. The semantic branch captures image content. The low-level branch preserves coarse outlines and structural detail. Quantitatively, the low-level branch performs better on structural fidelity metrics such as pixel correlation and SSIM, while the semantic branch leads perceptual measures. Combining them yields reconstructions that are both semantically accurate and structurally faithful, confirming their complementary contributions.

| Method | Low-Level Metrics | | | | High-Level Metrics | | | |
|---|---|---|---|---|---|---|---|---|
| | PixCorr ↑ | SSIM ↑ | Alex(2) ↑ | Alex(5) ↑ | Incep ↑ | CLIP ↑ | Eff ↓ | SwAV ↓ |
| Low level branch | **0.438** | **0.505** | 97.4% | 99.1% | 95.9% | 85.8% | 0.688 | 0.480 |
| Semantic branch | 0.336 | 0.431 | **98.5**% | 99.4% | 96.1% | 95.2% | 0.602 | 0.345 |
| Combined | 0.386 | 0.486 | 98.4% | **99.5%** | **97.3%** | **96.4%** | **0.564** | **0.320** |

Table T4: **Branch contributions.** Quantitative results for the semantic branch, the low-level branch, and the combined full pipeline. The low-level branch excels in structural fidelity metrics, while the full pipeline achieves a balanced performance across both semantic and structural metrics.

| Method | Low-Level | | | | High-Level | | | |
|---|---|---|---|---|---|---|---|---|
| | PixCorr ↑ | SSIM ↑ | Alex(2) ↑ | Alex(5) ↑ | Incep ↑ | CLIP ↑ | Eff ↓ | SwAV ↓ |
| BIT | **0.500** | 0.518 | **99.4%** | **99.8%** | **96.8%** | **87.5%** | **0.667** | **0.462** |
| MLP - InfoNCE loss | 0.267 | 0.481 | 80.9% | 85.0% | 77.5% | 69.1% | 0.897 | 0.613 |
| MLP - MSE loss | 0.182 | **0.520** | 54.2% | 54.2% | 52.0% | 51.1% | 0.976 | 0.670 |

Table T5: **Quantitative comparison between BIT and MLP in the low-level branch.** We assess the contribution of the Brain-Interaction Transformer (BIT) to low-level feature reconstruction by comparing it against a multilayer perceptron (MLP 1K hidden dim) baseline. The results show that BIT substantially outperforms the MLP predictor in reconstructing low-level visual features. Results are for Subject 1.

## B  Additional Quantitative Results

We present additional in-depth quantitative results. First, we report quantitative results on less saturated metrics than the ones shown in Sec. 5.2 (Table T6). Second, we present quantitative results on transfer learning reconstructions with less than 1 hour of subject-specific data (Fig. S3).

**Additional Metrics**  Many of the metrics commonly reported for fMRI image reconstruction appear saturated, suggesting little room for improvement and only small differences between methods. However, this impression is misleading, as it largely stems from the use of 2-way retrieval metrics and does not imply that the reconstruction problem is close to being fully solved. Conversely, some of the unsaturated standard metrics can also be misleading for other reasons. For example, the structural similarity index (Wang et al., 2004), when evaluated on grayscale images, fails to capture important chromatic information. To address this, we evaluate additional metrics that reveal substantial room for progress and highlight a significant performance gap between *Brain-IT* and other methods: (i) *1000-way retrieval* using CLIP embeddings of reconstructed versus ground-truth images; (ii) *LPIPS* (Zhang et al., 2018b) using AlexNet Features, reflecting perceptual similarity which correlates well with human judgment; (iii) *Color-SSIM-* evaluating SSIM on the RGB images.

| Method | SSIM-color ↑ | 1000way-CLIP ↑ | LPIPS ↓ |
|---|---|---|---|
| NeuroPictor | 0.375 | 0.255 | 0.628 |
| MindEye2 | 0.383 | 0.238 | 0.624 |
| MindTuner | 0.371 | 0.231 | 0.638 |
| **BrainIT(ours)** | **0.472** | **0.393** | **0.596** |

Table T6: **Additional quantitative metrics**. The additional metrics are evaluated on the reconstructions of subject 1, showing that when using non-saturated metrics, a big performance gap is revealed between Brain-IT and previous methods.

**Transfer learning on very limited data.**  In Table T7 we present quantitative results of reconstructions generated via transfer learning to subject 1 with varying amounts of subject-specific data—1 hour, 30 minutes, and 15 minutes . Complementing the qualitative results in Fig. S3, the quantitative evaluations demonstrate that *Brain-IT* effectively leverages its pretrained knowledge to adapt to a new subject, achieving strong performance even with as little as 15 minutes of subject-specific data (450 fMRI samples). To the best of our knowledge, this is the first time reconstructions from fMRI using less than one hour of subject-specific data are qualitatively and quantitatively comparable to those obtained with 40 hours of data in prior prominent methods.

| Method | Low-Level | | | | High-Level | | | |
|---|---|---|---|---|---|---|---|---|
| | PixCorr ↑ | SSIM ↑ | Alex(2) ↑ | Alex(5) ↑ | Incep ↑ | CLIP ↑ | Eff ↓ | SwAV ↓ |
| *BrainIT (15 min)* | 0.336 | 0.476 | 97.6% | 98.4% | 90.7% | 91.3% | 0.706 | 0.404 |
| *BrainIT (30 min)* | 0.378 | 0.480 | 99.1% | 99.3% | 94.3% | 93.3% | 0.652 | 0.372 |
| *BrainIT (1 hour)* | 0.396 | 0.484 | 99.4% | 99.6% | 95.9% | 94.7% | 0.619 | 0.348 |

Table T7: **Quantitative evaluations of transfer learning to subject 1 using different amounts of training data.** We show that even using 15 and 30 minutes of subject-specific data, Brain-IT produces reconstructions on par with previous methods of transfer learning.

## C   ADDITIONAL QUALITATIVE RESULTS

### C.1   RECONSTRUCTIONS WITH **40** HOURS OF DATA

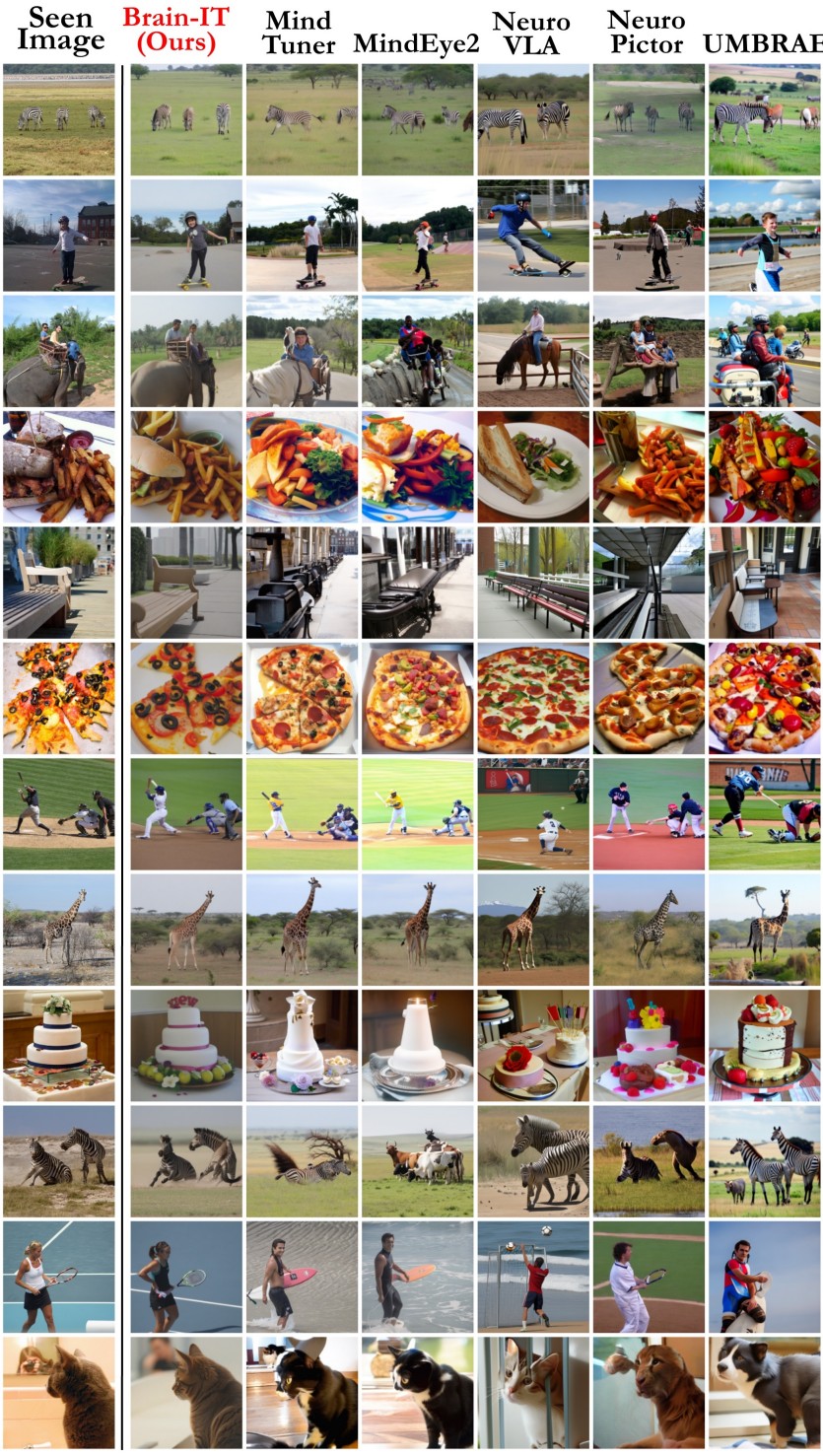

Figure S2: **Qualitative reconstruction results for subject 1.** Comparison to prominent previous methods on a large amount of images demonstrates the visual faithfulness of Brain-IT, alongside semantic accuracy.

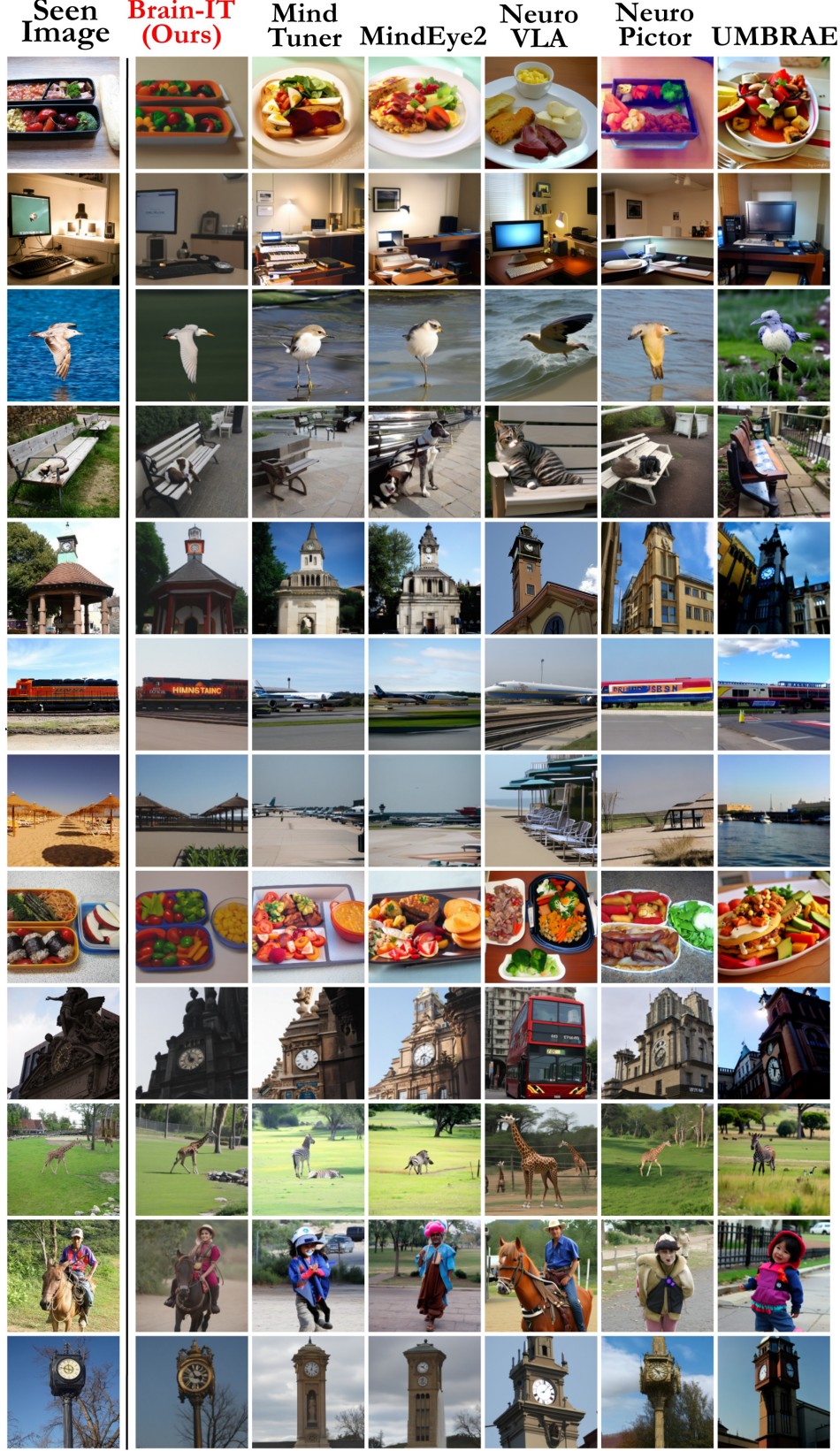

Figure S2: **Qualitative reconstruction results for subject 1**- Part B

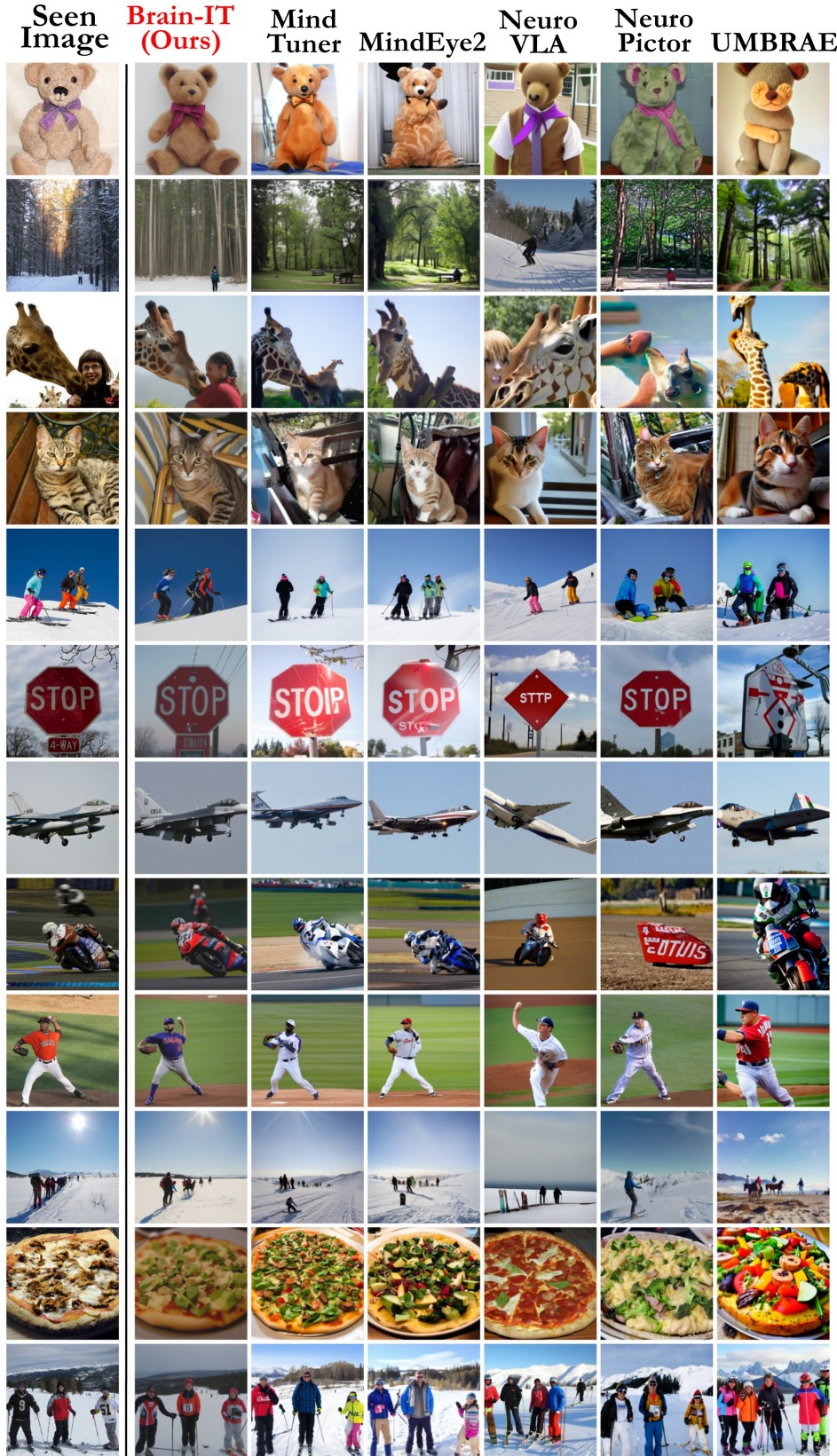

Figure S2: **Qualitative reconstruction results for subject 1**- Part C

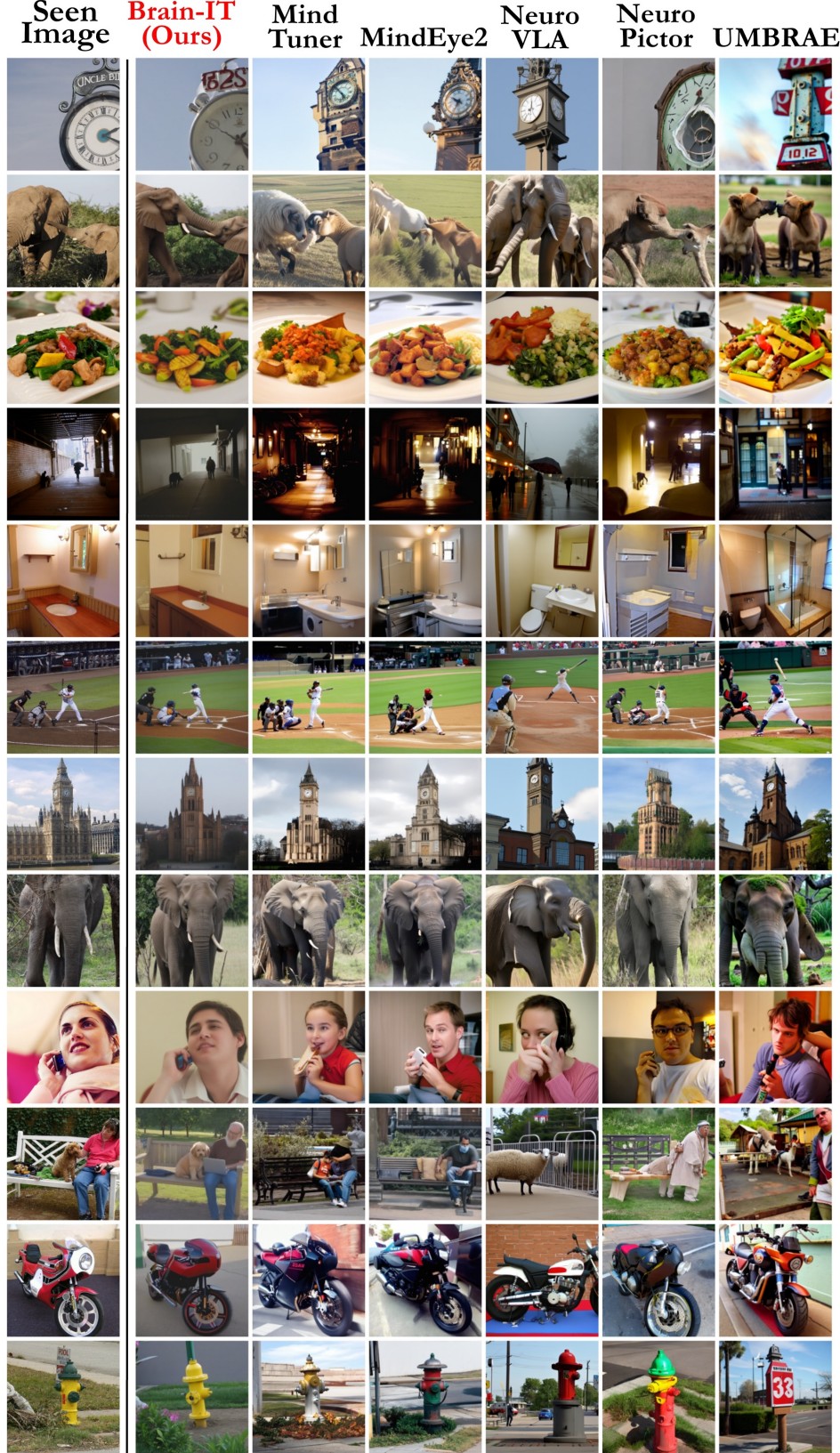

Figure S2: **Qualitative reconstruction results for subject 1**- Part D

## C.2 RECONSTRUCTIONS WITH TRANSFER-LEARNING ON LIMITED DATA

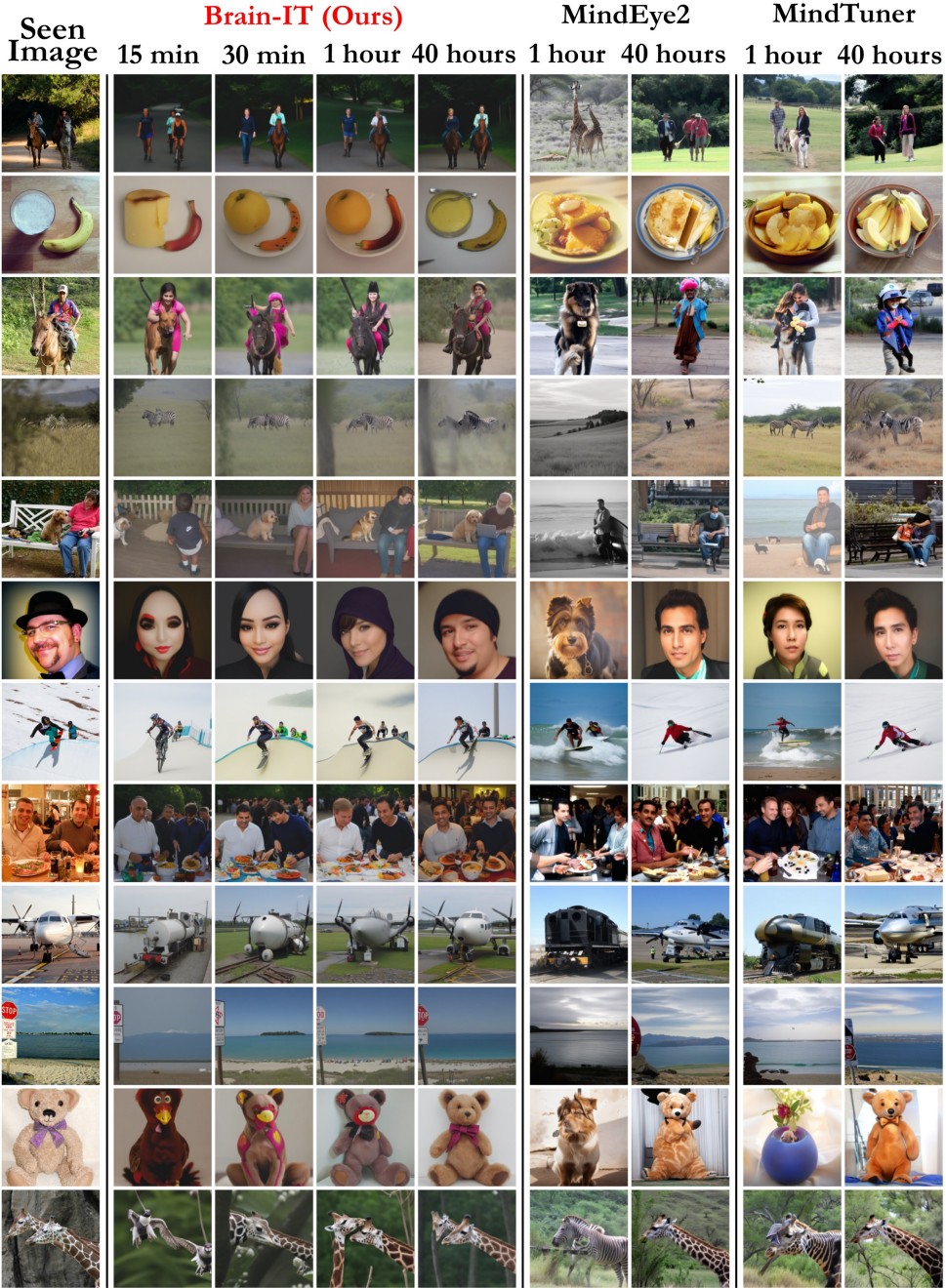

Figure S3: **Qualitative reconstruction results for transfer learning to subject 1 with limited data.** Brain-IT reconstructions using 1 hours and even 30 minutes are comparable to reconstructions of previous methods generated using 40 hours of data.

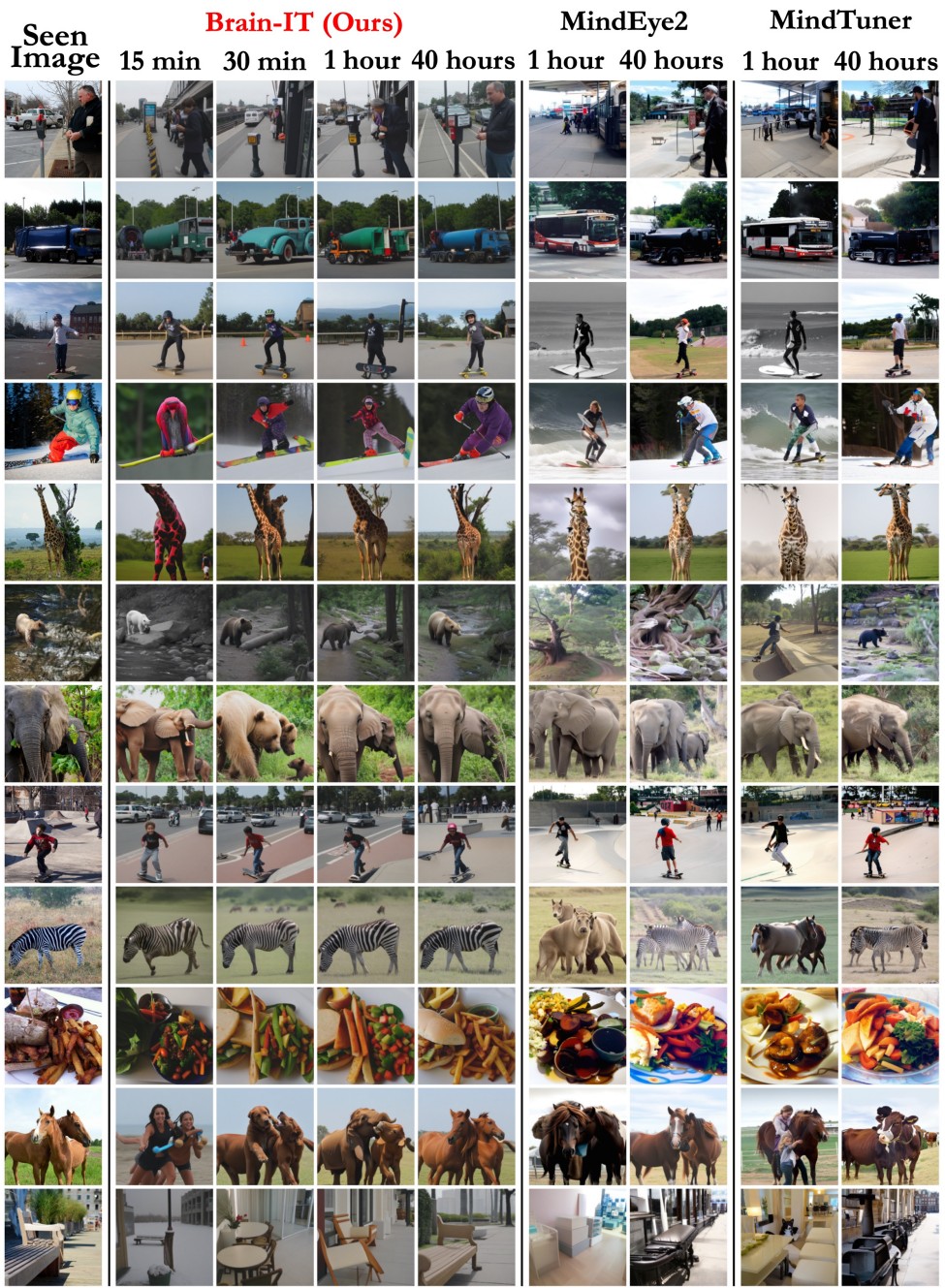

Figure S3: **Qualitative reconstruction results for transfer learning with limited data** - Part B

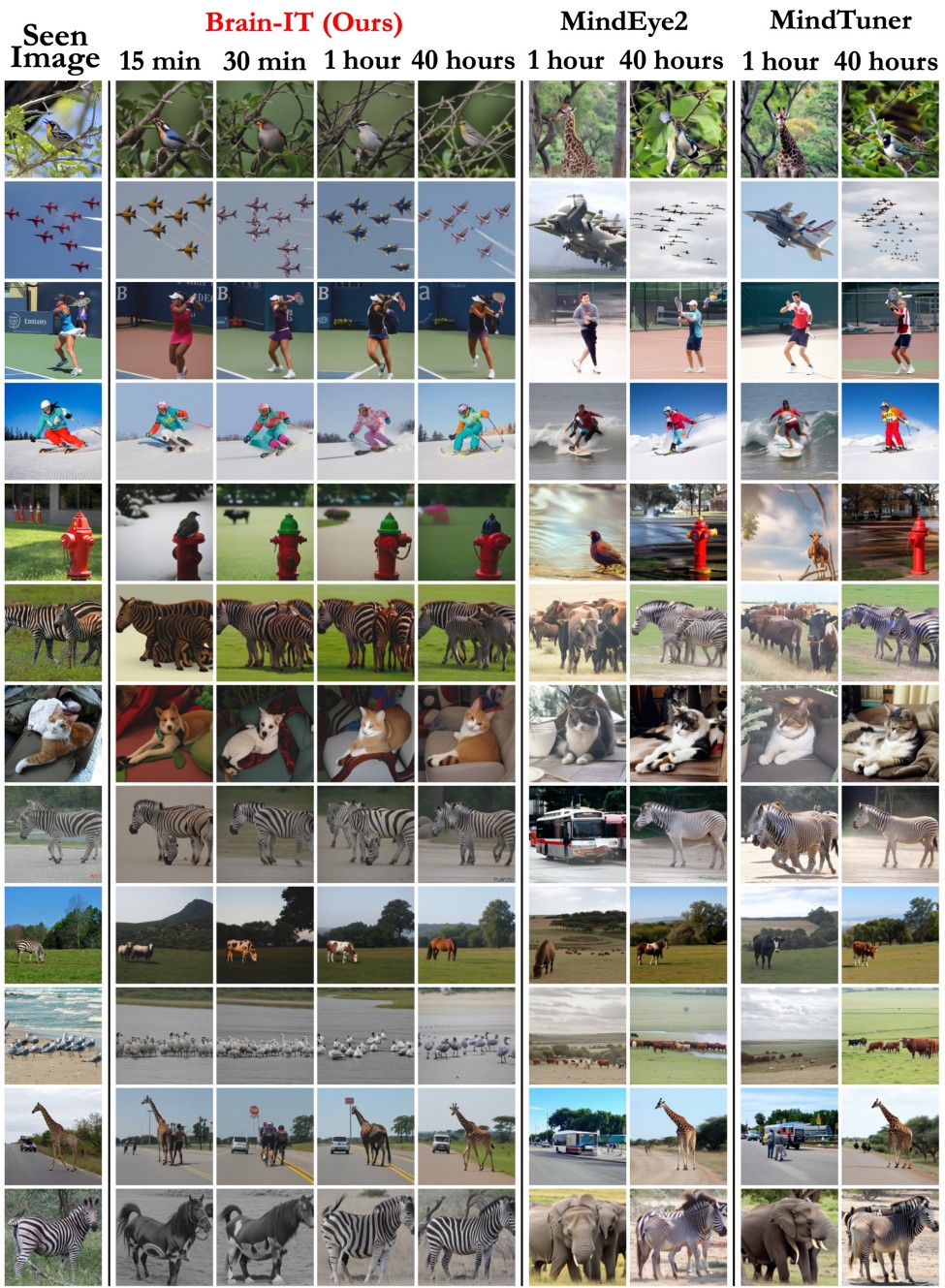

Figure S3: **Qualitative reconstruction results for transfer learning with limited data** - Part C

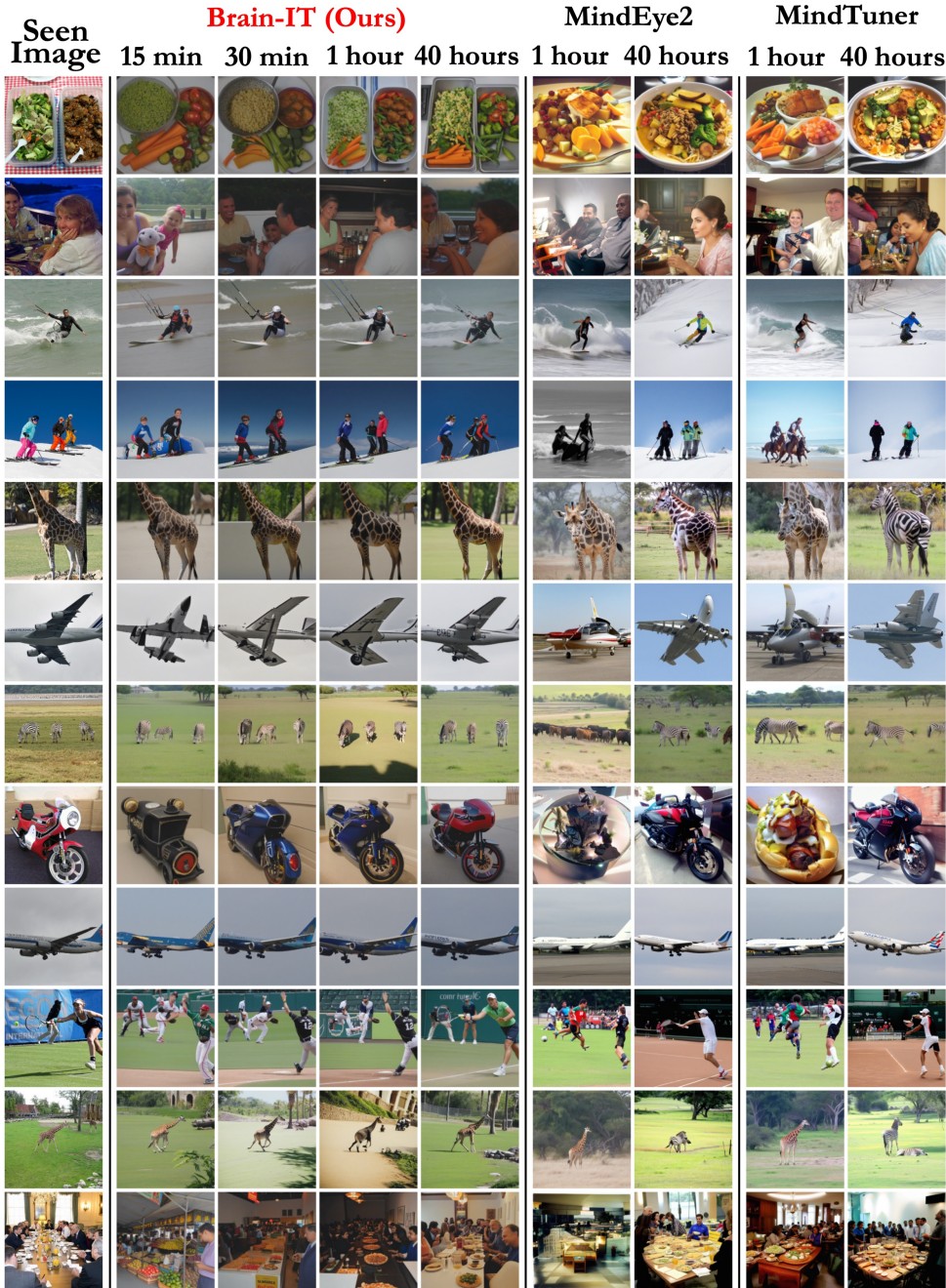

Figure S3: **Qualitative reconstruction results for transfer learning with limited data** - Part D

## C.3   IMAGE RECONSTRUCTIONS FOR ADDITIONAL SUBJECTS

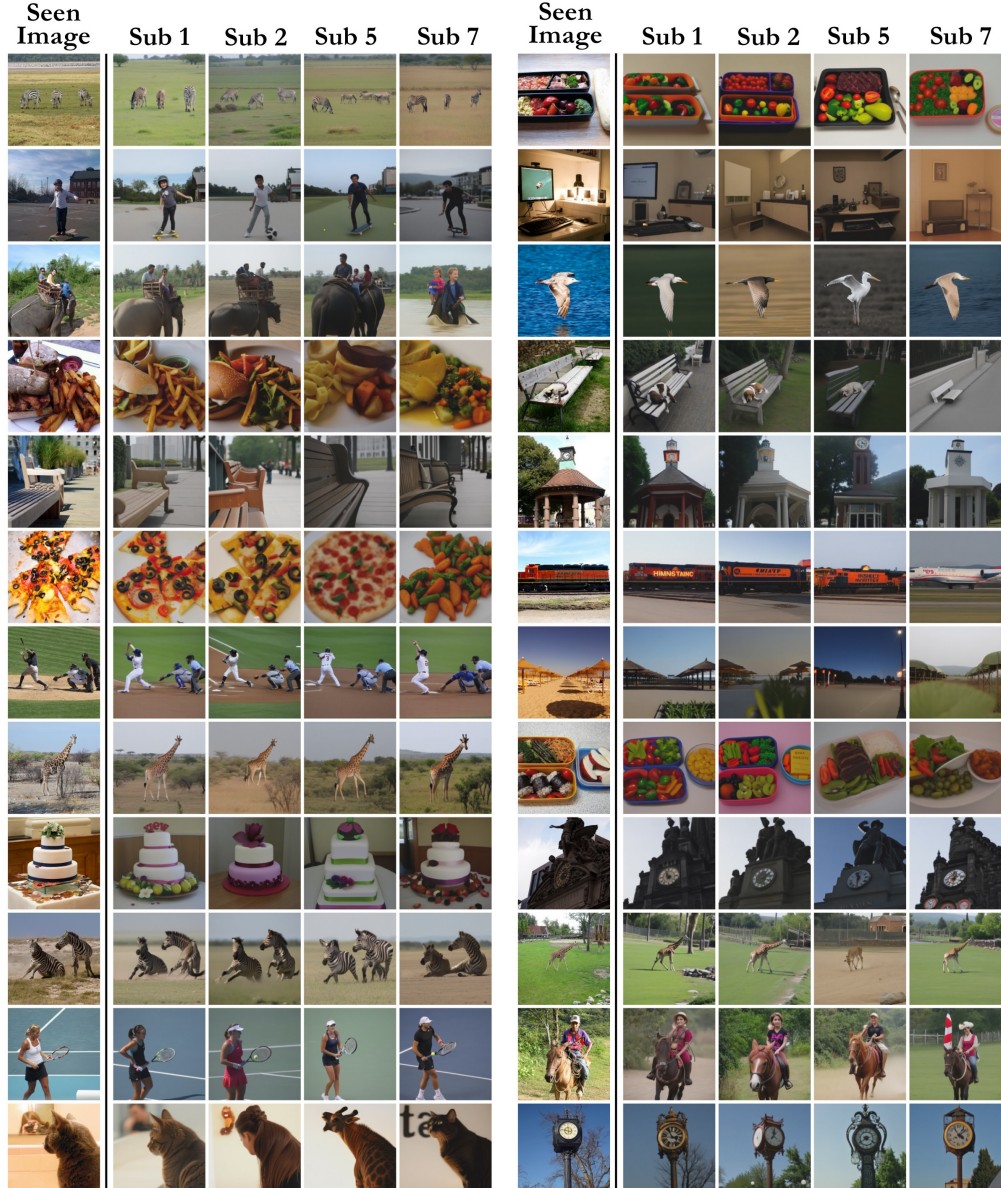

Figure S4: **Image reconstruction results for subjects 1, 2, 5 and 7.** Due to its use of shared functional clusters and network modules, Brain-IT successfully leverages cross-subject information to produce effective reconstructions for all four evaluated subjects

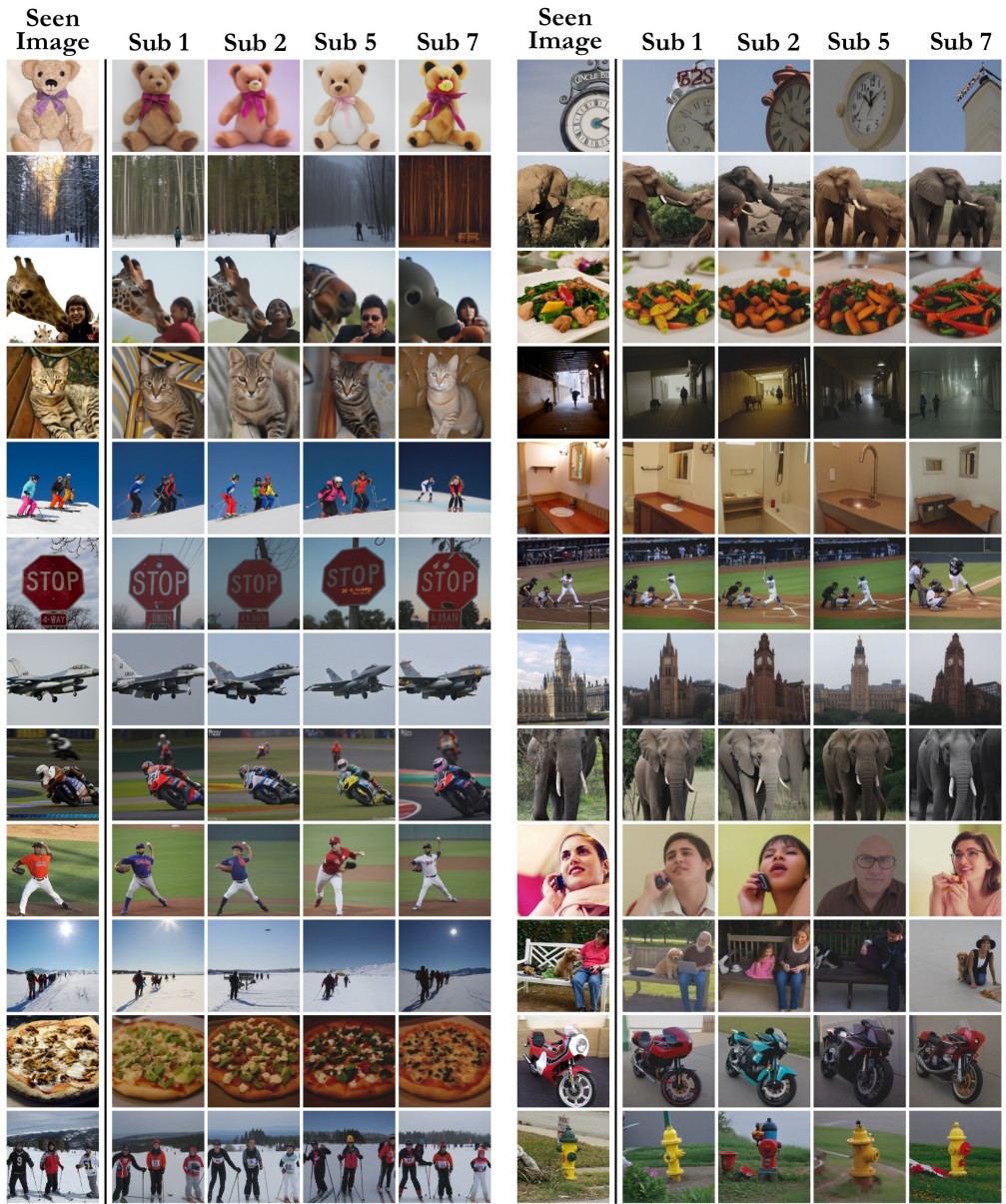

Figure S4: **Image reconstruction results for subjects 1, 2, 5 and 7** - Part B

## C.4 BRAIN TOKENS SPATIAL AND SEMANTIC SELECTIVITY

In Brain-IT, each fMRI voxel cluster is represented by a learned *brain token*, and each predicted image feature is associated with a learned *query token*. The brain tokens encode the functional activity patterns of specific voxel clusters, while the query tokens index the image features that the model must predict. These two sets of tokens interact through the cross-attention layers of the Brain-Interaction Transformer (BIT): for each query token, BIT attends over all brain tokens and aggregates their information to predict the image feature corresponding to that query.

Each query token is tied to a specific spatial position in an image feature map, so that every predicted feature can be localized in the image. Concretely, for the low-level branch, query tokens correspond to spatial positions in a VGG feature layer (an $A \times A$ grid), and for the semantic branch they correspond to spatial tokens in the CLIP image encoder (e.g., a $16 \times 16$ grid of patch tokens). Our goal is to show how different brain tokens contribute to the prediction of different image features via these cross-attention maps, thereby revealing the direct flow of information from functionally defined voxel clusters (brain tokens) to localized image features (query tokens) used for reconstruction. We present two complementary experiments: (i) brain-tokens that contribute to features at specific *spatial locations* in the image (see Fig. S5); and (ii) brain-tokens that contribute to features capturing specific *semantic concepts* in the image (see Fig. S6).

**Spatial selectivity of brain tokens.** In the first experiment, we visualize the cross-attention map between brain tokens and query tokens by averaging the attention weights across all transformer layers and all attention heads. For each brain token, this yields a 2D attention map over query tokens whose axes correspond to the spatial layout of the image features (an $56 \times 56$ grid for a VGG 1_2 layer, and a $16 \times 16$ grid of semantic spatial tokens). Each small square in Fig. S5a shows the attention map for one brain token; we arrange these squares to highlight tokens that consistently attend to specific image locations. In Fig. S5b, we assign a unique color to each brain token shown in Fig. S5a, and color all voxels in its corresponding cluster with the same color. This reveals which brain regions most strongly contribute to predicting features at particular image locations. As expected, we observe a clear contralateral organization: brain tokens in the right hemisphere predominantly support features in the left part of the image, and vice versa.

**Semantic selectivity of brain tokens.** In the second experiment, we demonstrate semantic selectivity rather than pure spatial position. For each brain token and each image, the corresponding fMRI response induces a different attention map from that token to all query tokens. We again average attention across layers and heads to obtain an attention map per brain token and per image, and then examine, across many images, which spatial regions (and thus which semantic content in those regions) a given brain token most strongly contributes to. We select brain tokens whose attention consistently highlights image regions with a clear semantic meaning, as determined by what typically appears in the attended parts of the images. Fig. S6 illustrates such semantically selective brain tokens, including tokens that preferentially contribute to predicting features for hands, legs, text/words, faces, and other object parts. For each example, we show both the attended image regions and the corresponding voxel cluster on the cortical surface, revealing where in the brain the voxel clusters that support these semantic predictions are located.

# Brain tokens spatial localization

## (a) Attention maps of brain tokens to query tokens

Each of the 6x6 maps corresponds to single brain token

VGG layer1_2                    Semantic

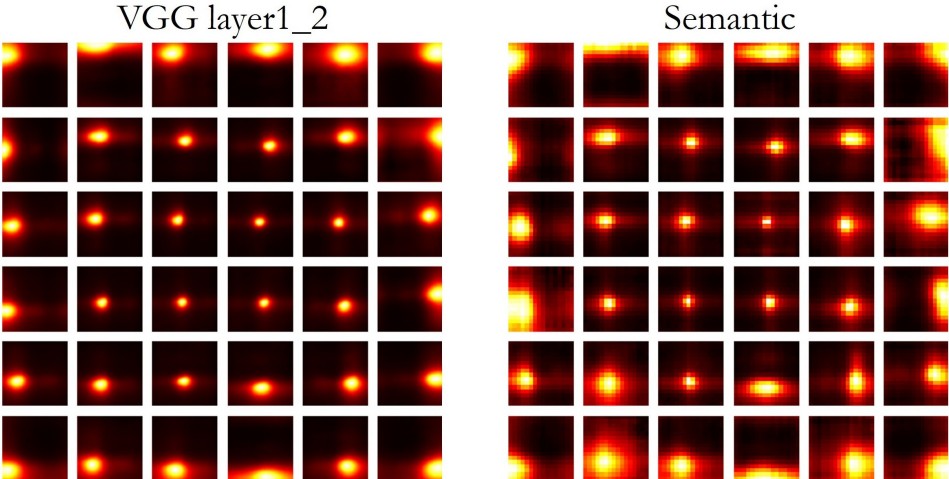

## (b) Voxel clusters assigned to brain tokens

Cluster color by
Attn-map position

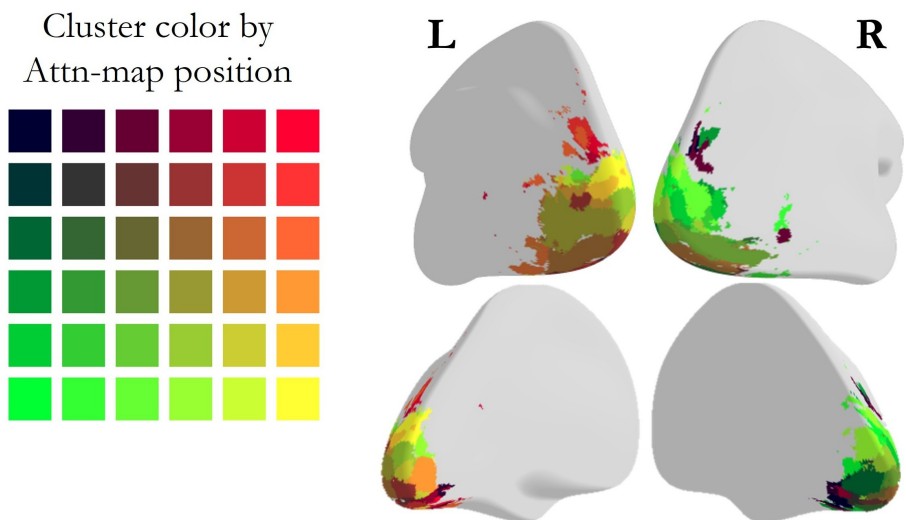

Figure S5: **Brain Token Spatial Attention Maps** (a) **Brain tokens' average attention maps** ($6 \times 6$ grid, 36 tokens shown), averaged across 1K test images. Each map demonstrates the spatial region in the reconstructed image that the corresponding brain token influences. (b) **Brain token brain maps**. A **color map** assigns a color to each brain token according to the position of it's attention map in the $6 \times 6$ grid (a). The brain map shows the voxels corresponding to each brain token, with their color determined by assigned color. This demonstrates the link between a token's **anatomical origin** and its **attended image location**.

## Brain tokens Semantic selectivity in BIT

### (a) Attention maps of brain tokens to query tokens

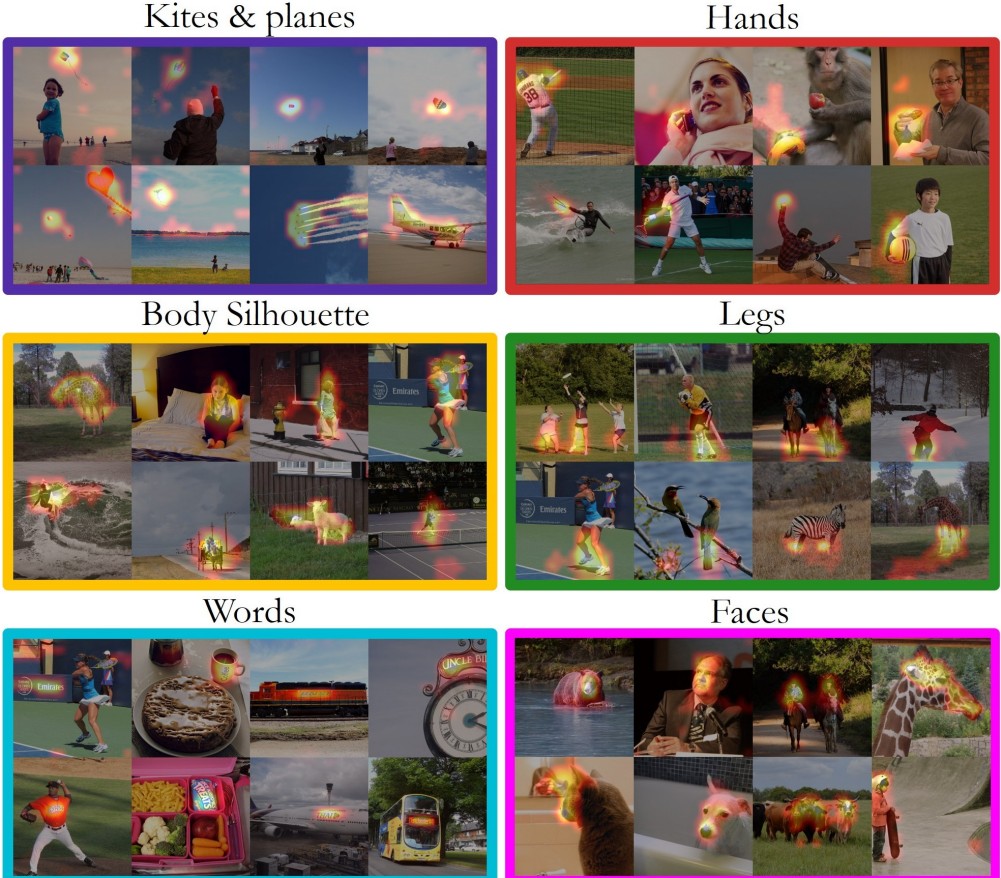

### (b) Voxel cluster assigned to brain tokens

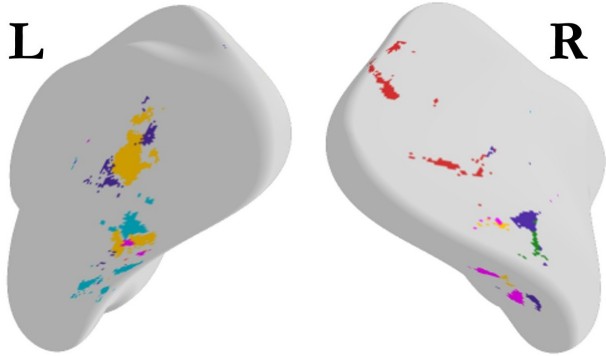

Figure S6: **Semantic Selectivity of Brain Tokens** (a) **Semantic attention maps**. Each image group showcases the average attention map of a single brain token within the Brain-IT semantic branch, overlaid on the original images. This is evaluated on the test images. We can observe clear semantic selectivity for each brain token (e.g., words, body silhouettes, faces). (b) **Brain token brain maps**. The brain map shows the anatomical source voxels of the tokens presented in (a), color-coded by the corresponding frame color in (a).

## C.5 RECONSTRUCTIONS ON NSD-SYNTHETIC

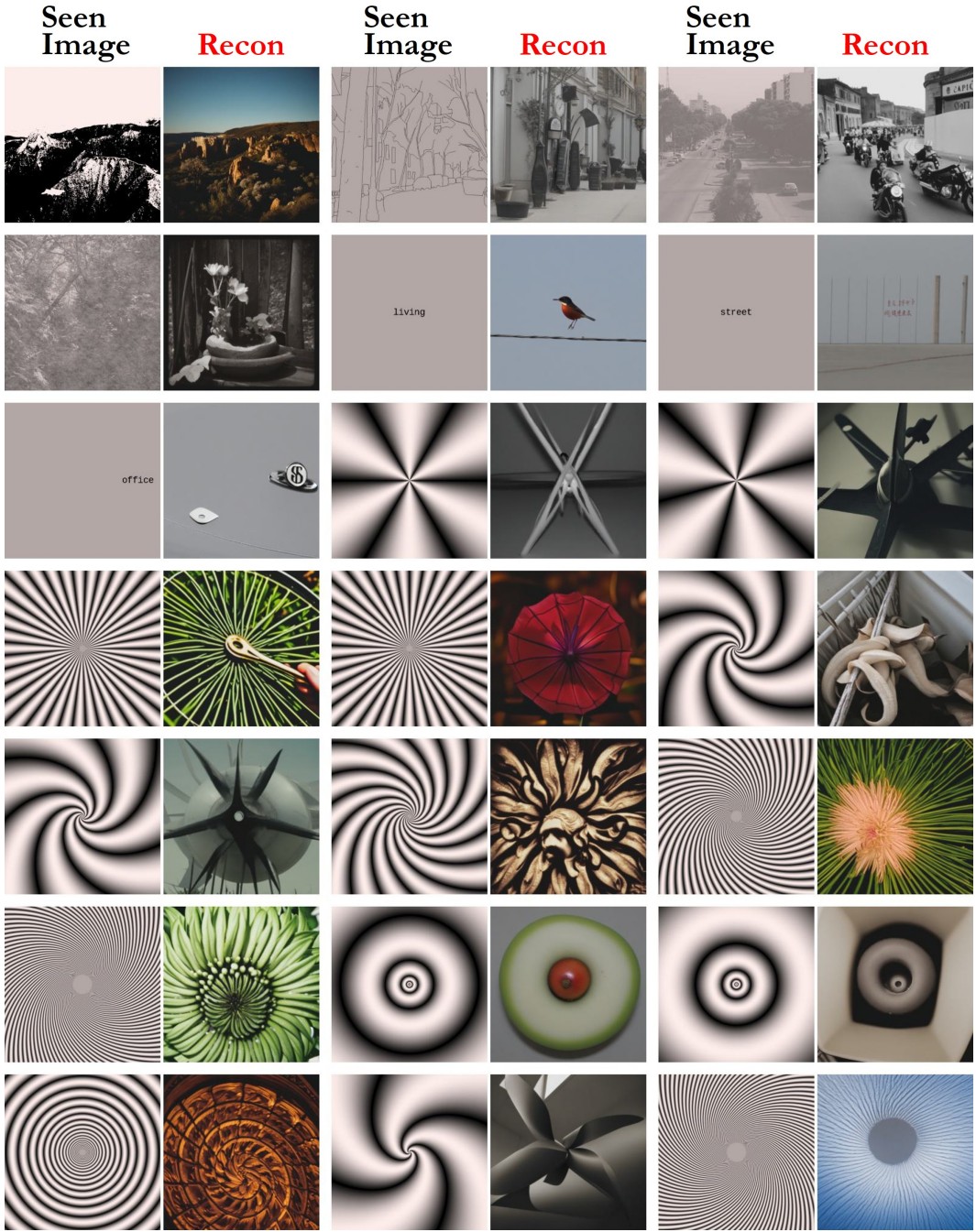

Figure S7: **NSD-Synthetic reconstructions** results for Subject 1 on the NSD-Synthetic dataset Gifford et al. (2024), an extension of NSD with out-of-distribution stimuli. Brain-IT is applied without any adaptation, demonstrating robust generalization to out-of-distribution stimuli.

## C.6 TWO-BRANCH CONTRIBUTION IMAGE RECONSTRUCTIONS

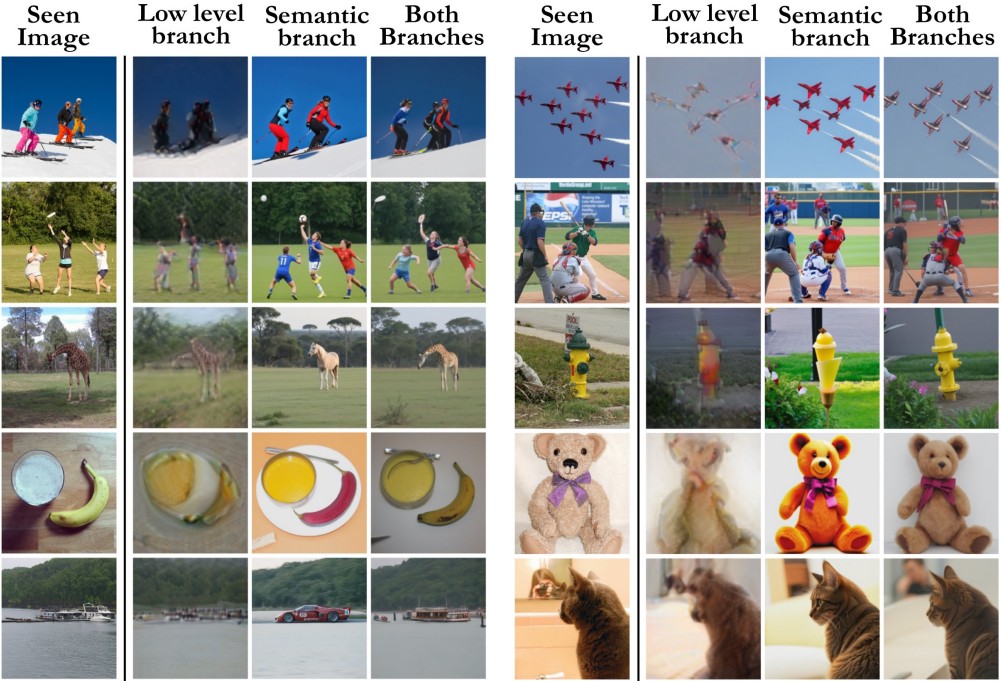

Figure S8: **Reconstructions via the two generation branches of Brain-IT**. The combination of both the low-level and the semantic branch ensures both semantically and visually accurate reconstructions. Importantly, the low-level branch often leads to accurate modifications of the semantic content, while the semantic branch often carries important visual information.

## C.7 FAILURE CASES

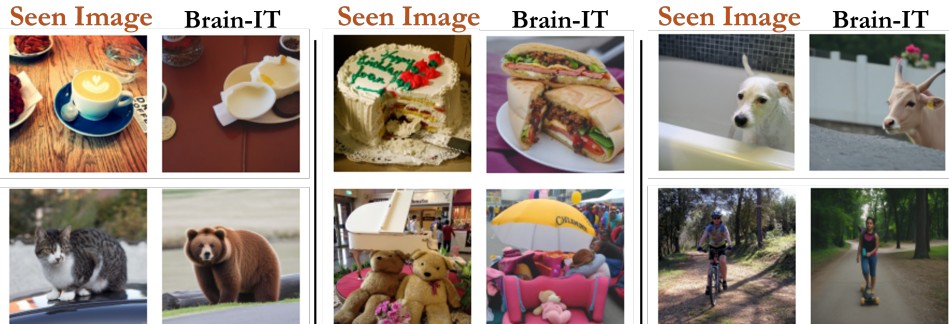

Figure S9: **Failure cases**. For certain fMRI samples, Brain-IT will predict either the semantic content or the visual layout of the image relatively accurately, but will fail to predict the other accurately, leading to incorrect reconstructions.

# D    ADDITIONAL TECHNICAL DETAILS

We provide additional details and explanations on training (Sec. D.1), the low-level branch (Sec. D.2), inference time generation (Sec. D.3), and transfer learning (Sec. D.4).

## D.1    TRAINING DETAILS

**Voxel sampling**    During training we sample voxels both to reduce memory usage and as a form of regularization. For each example, we uniformly sample 15K voxels with replacement from the $\sim$40K voxels of each subject. Our Brain Tokenizer, based on GNNs, can naturally handle a variable number of voxels, with the voxel-to-cluster mapping (GNN edge index) sampled consistently with the selected voxels.

**Enriching training with unlabeled images**    To augment training, we use $\sim$120k unlabeled images from the COCO unlabeled split. For each image, we predict synthetic fMRI using a Universal Encoder trained on the same training data, generating predictions for all eight subjects. During training, we iterate over all labeled NSD data and all unlabeled images in each epoch, and for each unlabeled image, we randomly sample the predicted fMRI of one subject as input.

**Training parameters**    We train the low-level branch (VGG prediction) using InfoNCE loss and the semantic branch stage 1 (CLIP alignment) using L2 loss. Both models are trained for 60 epochs using mixed-precision training and the AdamW optimizer with a learning rate of 5e-4 and a 15-epoch warmup. We apply ReduceLROnPlateau with a factor of 0.1 based on validation performance. The batch size is 64 for low-level and 128 for semantic (stage 1). We hold out 10% of the training set for validation to tune hyperparameters and select the best checkpoint.
Second stage of semantic branch (joint training )is trained with mixed precision using a batch size of 16 and gradient accumulation over 4 steps. We use the AdamW optimizer with a learning rate of 1e-5, and train for 10 epochs on images of size 256×256.

**Training times and resources**    Training times on H100 GPUs were: low-level branch 12 h (1 GPU); semantic branch stage 1: 8 h (1 GPU); stage 2: 10 h (4 GPUs).

## D.2    LOW-LEVEL BRANCH

Our low-level image reconstruction framework uses the DIP framework together with the VGG features predicted by the BIT model to produce a low-level image corresponding to the fMRI brain activity. This low-level image is then integrated into the diffusion process, resulting in reconstructions that are both semantically and structurally faithful to the actual seen image.

**VGG features**    We use a pretrained VGG-16 network with batch normalization to extract features. Features are computed from $112 \times 112$ images to reduce the number of positions to predict. We extract features from VGG layers 1_2, 2_2, 3_3, 4_3, and 5_3. For layers 1_2 and 2_2, we merge $2 \times 2$ adjacent positions into a single token by concatenation (1_2 without overlaps and 2_2 with overlaps), while for the remaining layers each position corresponds to a token. This results in $56^2$, $55^2$, $28^2$, $14^2$, and $7^2$ tokens for layers 1 through 5, respectively. The InfoNCE loss is applied to each layer separately. Different VGG layers have different numbers of channels, resulting in tokens of varying dimensionality. Tokens with fewer than 512 dimensions are replicated until reaching size 512. During training we randomly sample which positions (tokens) to predict for each layer. Specifically, we sample 512, 512, 128, 64, and 16 tokens from layers 1 through 5, respectively, in order to reduce memory usage. At inference time, we predict all tokens and reshape them back into convolutional layers, with overlapping positions (in layer 2) averaged.

**DIP inversion**    We use a DIP model based on a U-Net with input dimensionality 32, internal dimensionality 128, and 3 scales with bilinear interpolation. Training is performed for 2K iterations with initial noise of 0.1, regularization noise of $1/30$, and an exponential moving average with factor 0.99 applied to the output. All VGG layers are assigned equal weight during inversion. The inverted image has the same resolution as the VGG input ($112 \times 112$), and is upsampled to $256 \times 256$ to initialize the diffusion model.

### D.3 INFERENCE TIME GENERATION

To integrate the low-level image information with the semantic branch, we adopted an image-to-image configuration of the UnCLIP-based Stable Diffusion model. Instead of starting from pure Gaussian noise, we initialized the diffusion process with a partially noised version of the low-level branch output (the DIP inversion result), allowing the model to retain structural information while refining semantic content. The diffusion sampler performed 38 denoising steps in total, beginning from an intermediate noise level equivalent to step 14 in the noise schedule. Starting from this level adds a moderate amount of noise to the input latent, enough to enable semantic refinement, while still preserving much of the original structure.

### D.4 TRANSFER LEARNING TRAINING DETAILS

The transfer learning process consists of two steps: (i) We first adapt the "Brain Encoder" to the new subject as described in Beliy et al. (2024) (which can also be done with little training data). The resulting *Encoder* voxel embeddings are used both to assign each voxel to a cluster in the Voxels-to-Clusters (V2C) Mapping, and to initialize BIT's *Decoder* voxel embeddings. (ii) In the second step, we apply the ***Brain-IT*** training procedure to update the voxel embeddings of the new subject, while freezing all other components. Training is performed with the limited subject-specific samples, supplemented by predicted fMRI responses (on unlabeled data) from the adapted encoder in Step (i). This step mirrors the sequence of the original training pipeline (see Sec. 3) which include the two part: (i) Feature Alignment and (ii) Joint Training.

### D.5 EVALUATION AGAINST OTHER METHODS

We compare Brain-IT against several prominent fMRI-to-image reconstruction methods. For quantitative evaluation, we rely on previously published results reported by these methods. We follow the standard protocol adopted in prior NSD reconstruction papers: using the shared images seen by all subjects as the test set. Note that the full 40 sessions of NSD were released only after the completion of the 2023 Algonauts Challenge. Consequently, some methods were trained on 37 instead of 40 sessions (a relatively minor difference) and evaluated on 982 test images rather than the full 1000. MindEye2 recalculated evaluation metrics for several prominent baselines using their available code. All our experiments were trained on the full 40 sessions and evaluated on the complete test set. In our reported quantitative results, the following methods correspond to evaluations on 982 test images (rather than 1000): DREAM, UMBRAE, NeuroVLA, MindBridge and NeuroPictor. For qualitative comparisons and for computing additional metrics, we used reconstructed images provided by top-performing methods (some publicly available, others shared upon request).

