# OpenReview forum: "Brain-IT: Image Reconstruction from fMRI via Brain-Interaction Transformer"
_ICLR.cc/2026/Conference — ICLR 2026 Poster_

### Official Review · Reviewer_todj · 2025-10-29

**Soundness:** 3
**Presentation:** 3
**Contribution:** 3
**Rating:** 6
**Confidence:** 4

**Summary:**

This paper presents Brain-IT, a novel framework for reconstructing images from fMRI using a Brain Interaction Transformer (BIT). The method clusters functionally similar voxels across subjects to form shared functional units, which BIT maps to both semantic and structural image features for guiding a diffusion-based reconstruction. Experiments on the NSD show that Brain-IT achieves superior performance over existing methods in both qualitative and quantitative evaluations, while maintaining strong performance even with limited fMRI recordings.

**Strengths:**

* The brain-inspired BIT design is both biologically plausible and effective, leveraging shared functional voxel clusters to address data scarcity and inter-subject variability.
* The dual-branch reconstruction (structural and semantic) balances visual fidelity and semantic accuracy, producing reconstructions that better match the original stimuli.
* The efficient transfer learning strategy enables fast adaptation to new subjects with minimal fMRI data, greatly improving practicality.

**Weaknesses:**

* The paper lacks a clear distinction between its scientific and technical goals. While it emphasizes "brain interaction", it does not analyze what kinds of neural interactions BIT is meant to model or how these relate to known brain mechanisms, making the neuroscience contribution unclear.
* The Voxel-to-Cluster (V2C) mapping largely follows Beliy et al. (2024) but does not clearly explain what is new. How the shared clusters differ from prior work and how they are optimized for decoding rather than encoding?
* The low-level reconstruction choice (DIP+VGG) is insufficiently justified, with no comparison to standard alternatives like VAEs that could test whether the chosen method is actually superior.
* The anatomical clustering baseline is overly simplistic, relying only on 3D coordinates rather than established anatomical-functional parcellations (e.g., Schaefer et al., 2018). This makes the comparison less meaningful, as the performance gap may reflect a weak baseline rather than a true advantage of functional clustering.

[1] Schaefer et al. (2018). Local-Global Parcellation of the Human Cerebral Cortex from Intrinsic Functional Connectivity MRI.

**Questions:**

1. What specific scientific question does Brain-IT aim to address regarding brain function or visual processing?
2. How does BIT capture or validate specific neural interaction patterns between brain regions?
3. How does your V2C mapping differ technically and conceptually from Beliy et al. (2024), especially in achieving shared clusters?
4. Why was DIP+VGG chosen over VAE-based reconstruction, and have you tested this choice empirically?
5. Have you compared your functional clustering with established anatomical-functional parcellations such as Schaefer et al. (2018)? If not, why choose a coordinate-based baseline, and how might results change with a more realistic anatomical framework?

---

> ### Author Response · Authors · 2025-11-21
>
> **W1+Q2.** What kinds of neural interactions is BIT meant to model, or how do these relate to known brain mechanisms? How does BIT capture or validate specific neural interaction patterns between brain regions?
>
> *Answer:*
>
> We would like to clarify that BIT design is brain-inspired, reflecting general principles of distributed processing and interconnectivity in the brain. It enables interactions between groups of functionally related voxels, represented as Brain Tokens, through cross-attention, allowing information to flow across voxel clusters before being mapped to image features. While these learned interactions are designed to be biologically motivated, they are primarily intended to improve decoding rather than to directly model neural mechanisms.
>
> We have added new results showing what different brain tokens/voxel clusters capture in Appendix subsection C.4 (Figures **S5** and **S6**).
>
> **Q1.** What specific scientific question does Brain-IT aim to address regarding brain function or visual processing?
>
> *Answer:*
>
> Brain-IT addresses the problem of brain signal decoding, specifically the reconstruction of visual stimuli from fMRI signals.
> The problem of decoding is a core problem in the field of Neuroscience, addressing it will help us to answer important scientific questions (perception mechanisms, neural representations, semantic organization, etc.), and enables a variety of practical applications(Brain machine interface, clinical diagnostics, neurofeedback, etc. ).
> Performing thorough neuroscience analysis and making neuroscience claims is outside the scope of this paper. The paper focuses on the machine learning methodology of the problem and is justified primarily through a well-established and competitive benchmark.
>
> **W2+Q3.** How do the shared clusters differ from prior work and how are they optimized for decoding rather than encoding? How does your V2C mapping differ technically and conceptually from Beliy et al. (2024), especially in achieving shared clusters?
>
> *Answer:*
>
> Beliy et al. (2024) showed that voxel embeddings capture meaningful functional roles, which can be clustered across subjects. Building on these insights, we use the Beliy et al.’s embeddings to define a **functional parcellation**, which groups all brain-voxels of all subjects into 128 functional clusters. These clusters are formed as a **pre-processing step**, and are fed into our reconstruction model as meaningful building-blocks for our decoder.
>
> Brain-IT introduces an approach for training a decoding model on top of brain parcellation (shared clusters). Each such brain cluster is mapped to a new **learned brain-token**, which are used for self-attention across brain-clusters, and ultimately for Image Decoding.   BTW, the brain parcellation does not necessarily have to rely on Beliy et al.’s functional embeddings. Our framework can work with any brain parcellation (see answer to Q5 below).
>
> **W3+Q4.** The low-level reconstruction choice (DIP+VGG) is insufficiently justified. Why was DIP+VGG chosen over VAE-based reconstruction, and have you tested this choice empirically?
>
> *Answer:*
>
> Our choice of DIP+VGG is motivated by the following principles:
> - Alignment with brain representations: It was demonstrated that activations of networks trained on image classification align with brain responses to visual stimuli (Brain score Schrimpf et al.), making them good candidates for features predictable from fMRI signals.
> - Perceptual relevance: Multi-layer VGG features correlate with human perceptual judgments (LPIPS Zhang et al.).
> - Ease of inversion and robustness: Inverting VGG features is straightforward, allowing reliable reconstruction (style transfer Gatys et al.).
>
>  In our experiments, VGG provided more reliable reconstructions than newer Transformer-based models like MAE.
> Empirically, compared to prior works, including VAE-based approaches (e.g., Brain-Diffuser[Ozcelik et al.] employs VDVAE, MindEye[Scotti et al.] predicts the VAE of Stable Diffusion), our low-level reconstructions achieve better quantitative results, demonstrating the effectiveness of this representation.
>
> **W4+Q5.** The anatomical clustering baseline is overly simplistic. Have you compared your functional clustering with established anatomical-functional parcellations such as Schaefer et al. (2018)?
>
> *Answer:*
>
> Our framework can work with any parcellation; this stands in contrast to previous works that rely on anatomical grid patches. This flexibility stems from incorporating a graph neural network into our brain tokenizer. While we choose to work with a data-driven functional parcellation, following your request, we added results with the Schaefer 400 and 1000 region (7-network) parcellations in Appendix subsection A.2, **Table T2**. Our functional-based clustering performs better on most metrics, but the Schaefer-based parcellation is better than anatomical clustering baseline.

---

### Official Review · Reviewer_P5xc · 2025-10-30

**Soundness:** 2
**Presentation:** 3
**Contribution:** 3
**Rating:** 6
**Confidence:** 2

**Summary:**

The paper proposes Brain‑IT, a pipeline that maps fMRI signals to localized image features via a Brain‑Interaction Transformer (BIT) and reconstructs images through two complementary branches: a low‑level branch and a high‑level semantic branch. This design leverages coarse layout from the low‑level path to initialize diffusion and uses semantics to refine details. On NSD, the method achieves SOTA performance across most metrics, and demonstrates efficient transfer learning with only 15 minutes for new subject. The paper includes several ablations and states an intent to release code, checkpoints, and reconstructions.

**Strengths:**

- Brain‑IT outperforms prior methods on most of the conventional metrics in the 40‑hour setting and 1‑hour setting as well. It also reports first results for 15/30‑minute reconstruction on single subject.
- Ablations cover usage of external unlabeled images, functional vs. anatomical clustering, and number of clusters, plus branch‑wise contributions.
- They pledge to release code, checkpoints, and all reconstructed images upon publication.

**Weaknesses:**

- There is no explicit OOD evaluation (e.g., NSD‑synthetic) to probe robustness beyond the training distribution.
- The paper primarily focuses on reconstruction accuracy and does not analyze what the Brain Tokens capture, thus provides limited neuroscientific insights.

**Questions:**

- During pretraining on NSD, did the authors exclude the shared 1,000 images, i.e., are the shared images strictly held out throughout? I would like to confirm no leakage (or not).
- Related to the above question, I also would like to confirm transfer‑learning protocol and data usage.
    - For a new subject, which stages use that novel subject’s data, and how much? Please clarify if any of the new subject’s data is used in the shared pretraining stages or any other training stages.
    - For non‑target subjects (= training subjects), did the authors use all their data in all pretraining stages? Does this include or exclude the shared 1,000 images?
- For anatomical clustering, the paper states that voxels are clustered by 3D coordinates in FSaverage space and then run through the same pipeline. However, I wasn’t fully sure how the anatomical clustering baseline was implemented. Could the authors provide more detail on the procedure?

---

> ### Author Response · Authors · 2025-11-21
>
> **W1.** There is no explicit OOD evaluation (e.g., NSD‑synthetic) to probe robustness beyond the training distribution.
>
> *Answer:*
>
> Following your request we have added qualitative results for NSD‑synthetic Appendix subsectionC.5 **figure S7**.
>
> **W2.** The paper primarily focuses on reconstruction accuracy and does not analyze what the Brain Tokens capture, thus providing limited neuroscientific insights.
>
> *Answer:*
>
> Following your request, we provide an analysis of brain tokens, demonstrating their spatial localization and semantic selectivity. This is achieved by visualizing the average cross-attention maps between the brain tokens and the reconstructed image features (query tokens) in Appendix subsection C.4 (Figures **S5** and **S6**).
>
> **Q1.** During pretraining, did the authors exclude the shared 1,000 images, i.e., are the shared images strictly held out throughout?
>
> *Answer:*
>
> The 1000 shared images and their corresponding fMRI signals were strictly held out and not used in training at any stage, including during transfer learning.
>
> **Q2.** Clarification of transfer-learning protocol:
> - For a new subject, which stages use that novel subject’s data, and how much?
> - For non‑target subjects , did the authors use all their data in all pretraining stages?
>
> *Answer:*
>
> The transfer learning strategy includes the following stages:
> 1. **Pre-training on non-target subjects**: We create a subset of the data containing all the non-shared data of the 7 non-target subjects (excluding the shared test images). Next, we train an encoder (from Beliy et al., 2024) on this subset and use its embeddings to find a V2C mapping. Finally, we use this V2C mapping to train Brain-IT on this subset.
>
> 2. **Transfer learning**: To adapt the pre-trained models to our target subject, we use only the data from the first fMRI session (1 hour single trial fMRI recording) of their non-shared data. First, we train the non-target encoder on this subject-specific data and find a V2C mapping. Then, the voxel embedding portion of the Brain-IT model is finetuned on the subject-specific data (more details in App. D4)
>
> **Q3.** Clarification on anatomical clustering baseline implementation:
>
> *Answer:*
>
> For the anatomical clustering baseline, we use each voxel’s 3D coordinates (x, y, z) in fsaverage space, which is anatomically aligned across subjects. These coordinates are clustered using a Gaussian Mixture Model (GMM), following the same pipeline as the functional embeddings, effectively replacing the functional embedding with a spatial embedding. Additionally, we added evaluations for the anatomical-functional parcellation from Schaefer et al. (2018) in Appendix subsection **A.2** (Table 2). This demonstrates that Brain-IT is not dependent on a specific parcellation and can work with any desired parcellation.

---

> > ### Comment · Reviewer_P5xc · 2025-11-24
> >
> > Thank you for the answers and the additional analyses. Based on the authors’ responses, I am satisfied and have increased Soundness and Confidence.

---

### Official Review · Reviewer_zwfk · 2025-10-30

**Soundness:** 3
**Presentation:** 3
**Contribution:** 3
**Rating:** 6
**Confidence:** 5

**Summary:**

The paper introduces Brain-IT, a new framework for reconstructing natural images from fMRI data. Its key innovation is the Brain Interaction Transformer (BIT), which introduces functionally shared voxel clusters across subjects, mapped via a Gaussian Mixture Model over voxel embeddings derived from a pre-trained “Universal Brain Encoder.” Results show strong performance on both structural (PixCorr, SSIM) and perceptual (CLIP, Inception, AlexNet) metrics outperforming prior SoTA methods. Brain-IT sets a very strong benchmark for fMRI-Image decoding pipelines with SoTA performance in full and limited data settings. Brain-IT achieves comparable results to full-data baselines with only 1 hour of subject-specific data, and qualitatively plausible reconstructions even with 15 minutes.

**Strengths:**

1. **SoTA empirical results:** Outperforms all previous methods in full-data and limited-data settings.
2. **Novel techniques:** The idea of functional voxel clustering for fMRI-Image decoding is novel and improves performance, finding an elegant way to incorporate transformers into the mapper architecture.
3. **Extensive quantitative and qualitative evaluation:** The paper provides clear tables, broad baselines (MindEye, BrainDiffuser, MindTuner, MindEye2, NeuroVLA, etc.), and an ablation appendix exploring cluster number and anatomical vs functional clustering.
4. **Use of additional synthetic training data:** While the idea of using synthetic signals to improve alignment has been considered in previous works, using it to generate significantly more training data is a neat idea that seems to provide positive gains.

**Weaknesses:**

1. **Ambiguity in Functional Clustering Implementation:** While the voxel-to-cluster mapping is conceptually clear, I would appreciate a clarification on a few details:
- Are the voxel embeddings unique per-voxel-per-subject (i.e., one embedding vector per voxel that is fixed regardless of the input image), or do they vary per-image?
- Are these voxel embeddings frozen after the initial V2C mapping is established, or are they continuously optimized during BIT training? Section 5.3 mentions that adapting to a new subject requires optimizing only the voxel embeddings, suggesting they are trainable, but the timeline of when they are optimized versus frozen could be more explicit.
2. **Generalization to Out-of-Distribution Data:** The joint training approach in Stage (ii) of the semantic branch (Section 3.2) fine-tunes the diffusion model end-to-end with BIT on COCO-derived data. While you note this allows the models to "establish a representation that is better suited for conditioning," this differs from prior works that keep the diffusion model frozen. This raises a question about generalization: Could you comment on how this joint training might affect the model's ability to generalize to fMRI data from substantially different experimental paradigms (e.g., THINGS dataset with different stimuli distributions, or datasets with different scanning protocols)? While the strong NSD results are impressive, while pushing the SoTA on 7T fMRI results is impressive, the more important problem in the field is building models capable of transferring to new scanning protocols or modalities (M/EEG). Developing a method that gives gains on an already good baseline at the cost of increased overhead for cross-dataset generalization might undermine the contribution.
3. **Insufficient Evidence for "Localized Image Features" Claim:** Throughout the paper, you emphasize that BIT enables "direct flow of information from brain-voxel clusters to localized image features" as a key distinction from prior work. However, the predicted outputs are global CLIP embeddings (256 spatial tokens from ViT-bigG/14, which still represent the entire image) and VGG features. Could you provide evidence or analysis demonstrating that the functional clusters map to spatially localized regions in the predicted features? For instance, do specific clusters consistently predict features corresponding to particular spatial regions of images? Without such evidence, the "localized" claim appears to refer more to the intermediate representation (Brain Tokens) rather than the actual output features, which would make it similar to prior works in this regard.
4. **Lack of codebase:** I believe this is the biggest weakness of this work. I understand that the authors promised to release the code and checkpoints upon acceptance. But when you claim to have SoTA empirical results in a rapidly growing field, it is vital to provide the code necessary to validate your results. The gains are significant from previous works and without the ability to verify these gains, it does leave room for some skepticism.

**Questions:**

1. Verification of Low-Level Pipeline and VGG Model Details: The low-level reconstruction results are exceptionally strong with structural metrics (SSIM, PixCorr) being high while simultaneously maintaining strong semantic metrics (CLIP+Inception>85%), which is unusual since prior low-level pipelines typically show a trade-off (almost all previous works' LL pipelines have CLIP,Inception around the 50-60s). Given the novelty and importance of this component can you clarify the following:
- Which specific pretrained VGG checkpoint is used?
- Can you confirm there is no train/test overlap between the VGG pretraining data and COCO images used in NSD?

2. **Ablation on Low-Level Pipeline Contribution:** Another question regarding the low level pipeline. I believe your strong gains in the LL pipeline may be the primary driver of performance gains. Can you provide additional ablation experiments to isolate teh contribution of V2C+BIT from the VGG+DIP pipeline? For instance:
- Train an ME2 model on a single subject but switch the low level ground truth to VGG+DIP. Use an MLP similar to previous works feeding in the entire fMRI instead of doing V2C.
Anything along these lines to clarify whether the gains primarily come from the novel architectural choices (V2C mapping, BIT interactions) or the low-level reconstruction approach would be helpful.

3. **Loss Function Design Choices:**  You use the L2 loss for Feature Alignment and standard diffusion loss for Joint Training. This differs from recent works that employ contrastive losses, multiple alignment objectives, or diffusion priors to align fMRI with CLIP embeddings.
- Have you experimented with contrastive losses (e.g., InfoNCE) or other alignment objectives for the CLIP prediction task?
- What motivated using only L2 loss for the initial alignment stage?
- Could you report intermediate alignment metrics (e.g., cosine similarity between predicted and ground-truth CLIP embeddings) after each semantic training stage to better understand the alignment during the two stages?

4. Do you use image-to-text captioning (as in ME2) during the SDXL refinement stage, or only the CLIP-based conditioning?

Additional:
Typo in Figure 4: Trasnformer instead of Transformer


Overall I believe this a strong contribution to the field and I am open to increasing my score if my concerns are addressed.

---

> ### Author Response · Authors · 2025-11-21
>
> **W1.** Ambiguity in Functional Clustering Implementation:
> - Are the voxel embeddings frozen after the creation of the V2C mapping, or are they continuously optimized during training?
> - Are the voxel embeddings unique per-voxel-per-subject, or do they vary per-image?
>
> *Answer:*
> - Each voxel embedding is a learned per-voxel-per-subject representation, and doesn’t change between the images.
> - V2C mapping is a preprocessing step. Each voxel of each subject is assigned to a single cluster out of a total of 128 clusters. This is done before training, and doesn’t change during training.
>
> - There are two types of voxel embeddings: (i) the **decoding** voxel embeddings, learned by Brain-IT during training (both in the regular case, and in the Transfer-Learning);  and (ii) the **encoding** voxel embeddings, taken from [Beliy et al. 2024], which stay fixed throughout training. Importantly, both types of voxel embeddings are **unique per voxel per subject** and **do not vary across images**. We will clarify these issues in the revised version.
>
> **W2.** Generalization to Out-of-Distribution Data: Brain-IT finetunes the diffusion model in contrast to other methods that keep it frozen. How does this affect generalization to other datasets and possibly other modalities, and does this add increased overhead for cross-dataset generalization?
>
> *Answer:*
>
> Generalization is indeed a valid concern, and we believe it be a challenge for the majority of models. We can partially mitigate this in our framework through the use of synthetic fMRI data.
>
> Additionally, per the request of Reviewer P5xc, we have added reconstructions for “NSD-Synthetic” in Appendix C.5, **Figure S7**. “NSD-Synthetic” is an extension of NSD with Out-of-Distribution stimuli. We apply our model without any adaptation. These new results demonstrate the generalization capabilities and robustness of our model on out-of-distribution stimuli.
>
> **W3.** Insufficient Evidence for "Localized Image Features" Claim: Could you provide evidence or analysis demonstrating that the functional clusters map to spatially localized regions in the predicted features?  For instance, do specific clusters consistently predict features corresponding to particular spatial regions of images?
>
> *Answer:*
>
> Following your request, we visualize the average cross-attention maps between the brain tokens and the reconstructed image features (query tokens) in Appendix subsection C.4 (Figures **S5** and **S6**). These figures demonstrate the spatial localization and semantic selectivity of the brain tokens (functional clusters).
>
> **W4.** Lack of codebase: I understand that the authors promised to release the code and checkpoints upon acceptance. But, the gains are significant from previous works and without the ability to verify these gains, it does leave room for some skepticism.
>
> *Answer:*
>
> Due to our previous bad experience with releasing code prior to publication, we now (as a policy) do not release our research code before the paper is accepted. We plan to release our code upon acceptance.
>
> **Q1.** Verification of Low-Level Pipeline and VGG Model Details:
> - Which pretrained VGG checkpoint is used?
> - Can you confirm there is no train/test overlap between the VGG pretraining data and COCO images used in NSD?
>
> *Answer:*
>
> We used a VGG16-BN backbone, which employs ImageNet-pretrained weights and extracts features from specific layers.
> Checkpoint: torchvision.models.vgg16_bn(pretrained=True).
>
> There is no train/test overlap, since ImageNet and COCO are 2 disjoint datasets.
>
> **Q2.** Ablation on Low-Level Pipeline Contribution:  I believe your strong gains in the LL pipeline may be the primary driver of performance gains. Can you provide additional ablation experiments to isolate the contribution of V2C+BIT from the VGG+DIP pipeline?
>
> *Answer:*
>
> The performance of Brain-IT, even without the low-level branch, is very significant, as shown in Appendix Table T4. This shows that the performance gain is NOT only due to the low-level branch.
>
> Following your request we replaced BIT with an MLP to predict VGG features(with MSE  and InfoNCE losses), results provided in  Appendix **Table T5** (subsection A.5).

---

> > ### Author Response · Authors · 2025-11-21
> >
> > **Q3.** Loss Function Design Choices:
> > - Have you experimented with contrastive losses (e.g., InfoNCE) or other alignment objectives for the CLIP prediction task?
> > - What motivated using only L2 loss for the initial alignment stage?
> >
> > *Answer:*
> >
> > Our goal is decoding images from fMRI, predicting CLIP tokens is an intermediate step for that goal. Our guiding principle for connecting our model output directly to the conditioning of the diffusion model was simplicity. The first stage of training is aimed at aligning the BIT output with SD conditioning. We have experimented with MSE and InfoNCE objectives for this stage, and MSE performed better for us. In the second stage of end2end training BIT and the SD conditioning, BIT prediction diverges from CLIP embedding, and converges to a representation that is more suitable for reconstructing images from brain signals.
> >
> > **Q4.** Do you use image-to-text captioning (as in ME2) during the SDXL refinement stage, or only the CLIP-based conditioning?
> >
> > *Answer:*
> >
> > We do not use image-to-text captioning during the SDXL refinement stage; the only input to the model is the non-refined reconstructed image. We will clarify this in the revised version of the paper.

---

### Author Response · Authors · 2025-11-30

**Executive Summary of the rebuttal for the new AC:**

We would like to thank all reviewers for their valuable insights.
The reviewers found the approach promising and the results compelling — particularly,
the novelty of Brain-IT framework, and the strong results on the NSD benchmark.

The main concerns raised by the reviewers (and addressed in our rebuttal) involved:
**(i)** the biological interpretation of the model – specifically, they asked what our brain tokens represent,
and **(ii)** whether Brain-IT has generalization capabilities to out-of-distribution data.

To address these questions/concerns, we now added:

**(i)**  An analysis of spatial localization (receptive fields) and semantic selectivity of our brain tokens,
which are now added in **new Appendix C.4 (Figures S5 and S6)**. This new analysis demonstrates that our Brain-IT model learns meaningful representations suitable for informative analysis.

**(ii)**  Results on out-of-distribution (OOD) data – we now tested our Brain-IT decoder (without any adaptations) on a new extension of NSD called **"NSD-synthetic"**(OOD Test-data).
      These results of decoding fMRI of synthetic images were now added to **new Appendix C.5 (Figure S7)**,
      showcasing the robustness and generalization capabilities of Brain-IT to OOD.
      Please note that Brain-It was trained only on high-quality natural images,
      and never saw any synthetic images nor any poor-quality images in its training data.

---

### Meta-Review · Area_Chair_JCat · 2026-01-04

**Summary:**

This paper proposes Brain-IT, a brain-inspired framework for reconstructing images from fMRI using a Brain Interaction Transformer that operates over shared functional voxel clusters across subjects. Reviewers consistently found the approach novel and technically well designed, with particularly strong empirical performance on the NSD benchmark. The method achieves state-of-the-art reconstruction quality in both full-data and limited-data regimes, and demonstrates impressive transfer to new subjects with as little as one hour, and even minutes, of fMRI data. The main concerns raised by reviewers centered on biological interpretability, generalization beyond the training distribution, and implementation clarity rather than on empirical weakness. Overall, the paper presents a strong contribution at the intersection of machine learning and neuroscience.

**Reviewer Concerns:**

Most of the substantive reviewer concerns were addressed by the rebuttal. Questions about what the brain tokens represent and whether the claimed “localized image features” are meaningful were directly answered through new analyses visualizing spatial localization and semantic selectivity of brain tokens via cross-attention maps. Concerns about generalization and overfitting were alleviated by the added out-of-distribution evaluation on NSD-Synthetic, which demonstrates robustness without any model adaptation. Implementation ambiguities around voxel embeddings, functional clustering, transfer-learning protocol, and anatomical baselines were clarified in detail, and additional ablations were provided to isolate the contribution of the proposed architecture from the low-level reconstruction branch. One reviewer explicitly indicated satisfaction with these responses and increased their assessment accordingly.

Remaining concerns are relatively minor and mostly relate to scope rather than correctness, such as the limited neuroscientific claims, the decision to fine-tune the diffusion model, and the absence of public code at review time. These points were acknowledged by the authors, with reasonable justification and a clear commitment to release code and checkpoints upon acceptance. None of the remaining issues undermine the technical soundness or the significance of the contribution.

**Reviewer Scores:**

After reading the rebuttal and the ensuing discussion, my sense is that the reviewer who had initial but moderate reservations and actively engaged with the authors would likely have raised their score. The reviewers who were already on the positive side would probably keep roughly the same ratings, but with greater confidence following the additional analyses and clarifications. Importantly, no new technical issues emerged after the rebuttal, and nothing in the discussion suggests that any reviewer would lower their score. Taken together, the overall assessment either improves slightly or remains solidly above the acceptance threshold.

---

### Decision · Program_Chairs · 2026-01-26

Accept (Poster)